



# Public granaries as a source of proxy data on grain harvests and weather extremes for historical climatology

Rudolf Brázdil[1,2], Jan Lhoták[3], Kateřina Chromá[2], Dominik Collet[4,5], Petr Dobrovolný[1,2], Heli Huhtamaa[6,7]

[1]Institute of Geography, Masaryk University, Brno, Czech Republic
[2]Global Change Research Institute, Czech Academy of Sciences, Brno, Czech Republic
[3]Department of Historical Sciences, University of West Bohemia in Pilsen, Czech Republic
[4]Institute for Archaeology, Conservation and History, University of Oslo, Norway
[5]Center for Advanced Studies, Oslo, Norway
[6]Institute of History, University of Bern, Switzerland
[7]Oeschger Centre for Climate Change Research, University of Bern, Switzerland

*Correspondence to*: Rudolf Brázdil (brazdil@sci.muni.cz)

**Abstract.** Public granaries served as key infrastructures to improve food security in agrarian societies. Their history dates to the oldest complex societies, but they experienced a boom period during the 18th and early 19th centuries in Europe. In
Bohemia and Moravia (modern-day Czech Republic), numerous granaries were established by decree in 1788 to provide serfs with grain for sowing in the face of fluctuating weather. Here, we analyze granary data from 15 out of a total of 17 considered domains in the Sušice region (southwest Bohemia) from 1789 to 1849 CE. We use the documented annual values of grain borrowed by serfs, their grain depositions, total grain storage, and the total debt of serfs at the end of the year as proxies for harvest quality and size. Based on the series of these four variables, we calculate weighted grain indices,
considering the balance between borrowed and returned grain: a weighted bad harvest index (WBHI), a weighted good harvest index (WGHI), a weighted stored grain index (WSGI: WSGI-, more borrowed than returned; WSGI+, more returned than borrowed), and a weighted serf debt index (WSDI: WSDI+, more borrowed than returned grain; WSDI-, more returned than borrowed grain). WBHI, WSGI-, and WSDI+ were used to select years of extreme bad harvests, and WGHI, WSGI+, and WSDI- to identify years of extreme good harvests. We tested selected extreme harvest years against documentary
weather data and reconstructed temperature, precipitation, and drought series from the Czech Lands. We discuss the uncertainty of the data and the broader context of the results obtained. The findings document the potential of this new methodology, using widely available public granary data as proxies for historical climatological research.

## 1 Introduction

Past agrarian societies were vulnerable to weather and climatic patterns that influenced the annual agricultural cycle (Brázdil
et al., 2019b), which were reflected in the quantity and quality of agricultural production (e.g., Jones, 1964; Brunt, 2015; White et al., 2018a; Skoglund, 2023; Martínez-Gonzáles, 2024). The production of grain, the main source of caloric energy



in Europe, was particularly crucial for food security. Any deficit in grain production had substantial knock-on effects on the wider economy (e.g., Wrigley, 1989; Edvinsson, 2012; Esper et al., 2017; Rácz, 2020). Due to limitations in trade and transport, the impacts of harvest fluctuations were often local and regional. In combination with economic, societal, and
political stress, shortfalls could potentially result in social unrest or even famines (e.g., Campbell, 2009; Campbell and Ó Gráda, 2011; Brázdil et al., 2018; Huhtamaa, 2018). Adverse weather and its extremes could affect the entire cereal growing cycle, from sowing to harvest (e.g., Petr, 1987; Brázdil et al., 2019b; Trnka et al., 2023; Skoglund, 2024). Extended wet or dry periods during autumn could negatively influence the sowing of winter crops (wheat, rye), while extreme winter and late spring frosts could cause frost damage, and mild, overly wet conditions could lead to rotting crops. Spring crops (barley,
oats) were vulnerable to prolonged snow cover, frosts, and cold temperatures. Both overly wet or dry conditions could complicate their sowing, growing, blossoming, and ripening. Furthermore, wet conditions during the grain harvest could impede and delay fieldwork, while also increasing potential losses during storage (e.g., Petr, 1987; Eitzinger et al., 2013).

The quantity and quality of grain harvests were especially reflected in fluctuations of grain prices, despite many other socio-economic and political factors influencing their final value (e.g., Petráň, 1977; Scott et al., 1998; Ljungqvist and Seim,
2024). Generally, good grain harvests were followed by lower prices, and poor harvests by higher prices. The complex relationship between weather/climate, harvests, and grain prices has long been studied, primarily in local, regional, or national settings (e.g., Pfister, 1988; Hildebrandt and Gudd, 1991; Bauernfeind and Woitek, 1999; Brázdil and Durďáková, 2000; Holopainen et al., 2012; Camenisch, 2015; Huhtamaa et al., 2015; Yin et al., 2015; Pribyl, 2017; Skoglund, 2022; Brázdil et al., 2024). In recent years, several studies have explored this connection on a broader territorial scale. These
studies investigated European cases, examining environmental drivers of fluctuations in historical grain prices (Esper et al., 2017), climate variability and grain prices (Ljungqvist et al., 2022), and climatic signatures in grain harvest yields (Ljungqvist et al., 2023). Moreover, the latter study emphasized the importance of research analyzing the relationships between grain quantity and quality, their links to climate, and the subsequent effects on grain prices in historical contexts.

As reduced or failed harvests affected large parts of society, measures to avoid shortages and famine became a crucial part of
administrative planning. The establishment of granaries had been a popular instrument for safeguarding against fluctuations since antiquity (e.g., Morales et al., 2014; Salido Domínguez, 2016). In the early modern period, these security infrastructures expanded in line with the ambitions of administrations. They addressed not only food security issues but also supplied the military and growing cities, while providing governments with some control over the crucial food market. As such, they linked ecological factors with military, economic, and social concerns (Collet, 2010). Some polities, such as the
Inka Empire, early modern China, or 18th-century Prussia, can be understood as veritable "granary states" centered around staple economies (e.g., Liu and Fei, 1978; Will and Wong, 1991; Polanyi, 2018; Shiue, 2024). In Europe, public granaries rose in prominence throughout the 18th century (see, e.g., Franko, 1907 for former Galicia – now part of Poland and Western Ukraine; Černý, 1932 for the Czech Lands; Kaplan, 1977 for France; Teerijoki, 1993 for Finland; Berg, 2007 for Sweden; Collet, 2010 for Prussia; Seppel, 2019 for Russia; Løvdal, 2020 for Norway). Eventually, improved transportation during the





late 19th and early 20th centuries allowed grain storage to be moved out of cities, dispersed, and gradually privatized (Collet, 2010).

So far, public granaries in Europe have been studied from historical, socio-economic, technological, and political perspectives. This paper, however, focuses on the use of granary records as proxies for the study of poor and abundant grain harvests and their relationships to weather/climatic factors. Based on public granary data from southwest Bohemia (modern-

day Czech Republic) for the period 1789–1849 CE, we argue that such widely available data provide a valuable new source of information for historical climatological research.

## 2 Data

### 2.1 The regional setting

The town of Sušice in southwest Bohemia was used for a detailed analysis of the relationships between grain prices and

weather/climate effects during the period 1725–1824 CE by Brázdil et al. (2024). Due to the rich regional record of historical and documentary weather/climate evidence, a broader area around this town was selected as the target region for analyzing grain data from public granaries (Fig. 1). The excerpted data from the granaries pertain to a total of 17 domains (Table 1), representing 168 settlements or their parts and 30,305 inhabitants, covering approximately 448.87 km² of area (the smallest being 1.44 km² for Lipová Lhota and the largest 101.31 km² for Žichovice). This region lies within the broader foothills of

the Šumava Mountains, spanning a wide altitudinal range from *c*. 440–1140 m asl. According to the Köppen climatic classification, this region falls under the Cfb type – temperate broadleaf deciduous forest, with mean annual temperatures *c*. 6–8 °C and mean annual precipitation totals of *c*. 550–700 mm (Tolasz et al., 2007). In terms of grain production, the main cereals grown in the studied area and period were rye and oats, while barley and particularly wheat were cultivated to a lesser extent.



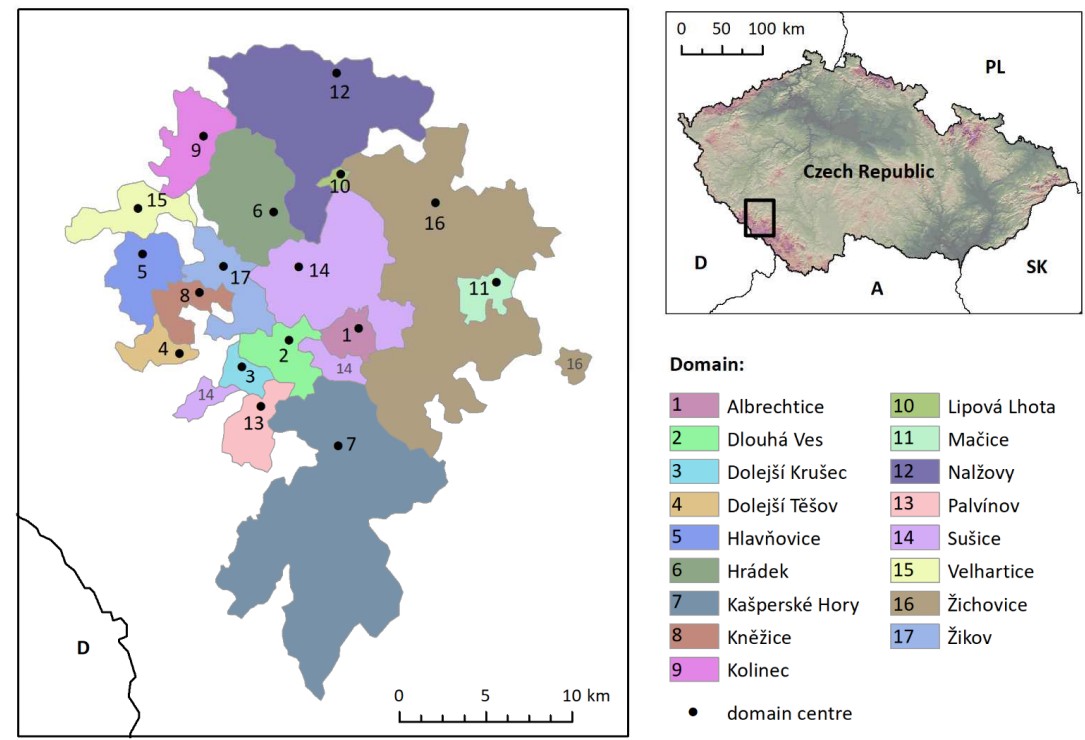

**Figure 1.** The investigated Sušice region in southwest Bohemia, consisting of 17 individual domains with available public granary data, and its position within the Czech Republic. Abbreviations: A – Austria, D – Germany, PL – Poland, SK – Slovakia.

**Table 1.** Basic information about the analyzed domains in the Sušice region of southwest Bohemia from the 1830s, according to Sommer (1839, 1840). The altitude range represents the altitudinal interval of the corresponding villages in each domain; the number of villages also includes their parts when a village belonged to two domains; x – the grain fund of the domain was not created.

| Name | Area (km²) | Altitude range (m) | Number of villages | Number of inhabitants | Granary records | Archival source |
|------|-----------|--------------------|--------------------|-----------------------|-----------------|-----------------|
| Albrechtice | 6.58 | 504–714 | 5 | 393 | 1794–1845 | AS8 |
| Dlouhá Ves | 8.42 | 512–600 | 3 | 904 | 1792–1849 | AS13 |
| Dolejší Krušec | 3.52 | 621–657 | 4 | 404 | x | AS19 |
| Dolejší Těšov | 4.89 | 737–876 | 4 | 316 | 1808–1839 | AS20 |
| Hlavňovice | 12.12 | 610–790 | 10 | 871 | 1792–1848 | AS10 |
| Hrádek | 27.53 | 485–793 | 10 | 989 | 1789–1849 | AS11 |





| Kašperské Hory | 97.32 | 563–1144 | 26 | 4593 | 1810–1849 | AS9 |
| Kněžice | 7.43 | 547–820 | 6 | 612 | 1805–1849 | AS12 |
| Kolinec | 17.74 | 545–737 | 5 | 1472 | 1789–1849 | AS7 |
| Lipová Lhota | 1.44 | 555 | 1 | 74 | x | AS14 |
| Mačice | 6.80 | 567–652 | 3 | 583 | 1794–1849 | AS15 |
| Nalžovy | 68.14 | 458–584 | 20 | 4941 | 1789–1848 | AS6 |
| Palvínov | 12.44 | 520–770 | 10 | 954 | 1792–1849 | AS16 |
| Sušice | 46.18 | 465–778 | 14 | 3327 | 1786–1844 | AS18 |
| Velhartice | 12.24 | 622–705 | 4 | 651 | 1791–1849 | AS21 |
| Žichovice | 101.31 | 442–1014 | 30 | 8185 | 1789–1845 | AS17 |
| Žikov | 14.77 | 504–737 | 13 | 1036 | 1816–1845 | AS22 |

### 2.2 Public granary data

Although public granaries (*kontribuční sýpky*) already existed in the Czech Lands earlier (Vodnařík, 1886; Krofta, 1949),
their systematic establishment in the Kingdom of Bohemia and the Moravian Margraviate followed the official patent no.
241 issued by Emperor Josef II on 9 June 1788 (Kalousek, 1910; Figure 2a). This decree was part of a broader European
discussion on these infrastructures, which emerged in response to the climatic shocks of the 1770s and 1780s and connected
to the emperor's "grand plan" to reform agriculture (Pfister and Brázdil, 2006; Collet, 2019). However, the main purpose of
these new granaries was no longer general food security or supplying the military; instead, they specifically aimed to secure
100    seed grain for serfs, a focus that significantly reduced the impact of political interference. The emperor drew upon existing
voluntary systems at some domains to mandate the establishment of obligatory so-called crop funds at each domain from 1
November 1788. This support for serfs was driven more by concerns for the stability of the empire's tax base than by
humanitarian goals (Černý, 1932).

In theory, each peasant was annually required to transfer an amount equal to one-third of the seed grain needed for each type
105    of cereal. In case of damage caused by a natural disaster, peasants could borrow grain from the crop fund created by these
deposits, with the obligation to return it after the next harvest along with an interest rate of 12.5 %. These transactions were
documented in special accounts attached to the regular granary reports for each domain. These reports were sent to the
regional office and then to the Czech government (*Gubernium*), where they have usually been preserved to the present day.

The reality often looked somewhat different. Granaries were dynamic institutions where a range of interests overlapped and
110    competed. The establishment of the crop funds often took several years to implement. The serfs had to cover the building
costs, some estate owners were slow to respond to the decree, and the required amounts of surplus grain were hard to come
by (Prasek, 1904). In addition, the intended volume of annual deposits and the level of interest proved unrealistic. Already in
1793 CE, deposits were reduced to one-twentieth of the required seed grain, and the interest for loans was dropped to 6.25 %





(in Moravia, interest was decreased to the same level as late as 1812). This was in response to the corn funds remaining

incomplete for years, being loaned out in full, or only existing on paper in the first place. The hardship brought by the Napoleonic wars exacerbated this problem. The Czech *Gubernium* intensified oversight and reminded overseers to require the prompt return of loans to avoid desperate serfs falling into spiraling debt. Later, it emphasized lending grain only to dependable applicants and requested constant updates on the movement of crop funds. It also asked for relief on interest payments to be documented individually at the level of individual domains. Not least, the fluid agricultural ecology often

corresponded imperfectly with administrative requirements, as late harvests and threshing dates into November could jeopardize annual deposits (for example, in Albrechtice – archival source AS8).

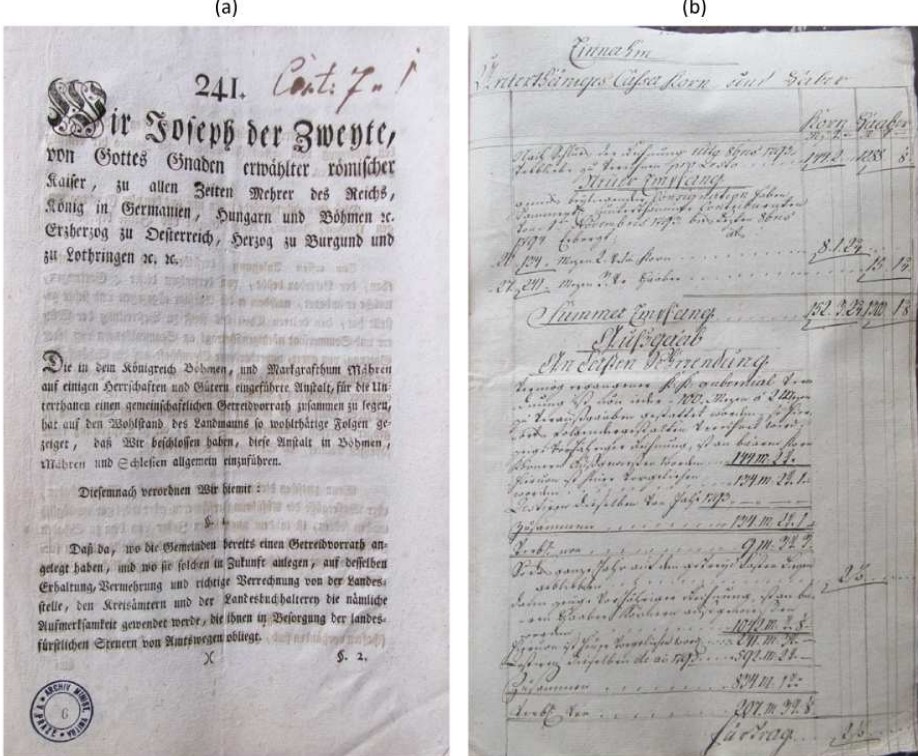

**Figure 2. (a) Title page of patent no. 241 issued by Emperor Josef II for the establishment of public granaries in the Czech Lands (Národní archiv, České gubernium - contributionale, karton č. 1580), and (b) an example of granary records from the Hrádek**
**domain (AS11, karton č. 1586).**

Cereal cultivation and record keeping often remained mismatched. Rye and oats dominated the granary records in southwest Bohemia, while barley was reported for six domains, and wheat only for the Mačice domain and partly for the Žichovice





domain. Granary accounts for these crops were kept according to the military year (from 1 November of the preceding year until 31 October of the current year), and measurements were recorded in Lower-Austrian units: *měřice* (1 *měřice* = 61.49

L), *věrtel* (1 *věrtel* = 15.37 L), and *měřička* (1 *měřička* = 3.84 L). During this time, interest was added to borrowed grain, while annual storage losses (due to pests, humidity, technical faults at the granary, etc.) were subtracted. The resulting quantities of borrowed and returned grain were recorded in account supplements, which listed the total loans and interest due from serfs. Together with the quantity of grain present in the granary, these figures were entered into the accounts as the value of cereal at the end of the military year. In hindsight, it is precisely because rather than in spite of these practical

challenges that the granary records are so rich and valuable today.

### 2.3 Weather, climate and harvest data

Weather and its extremes were taken from the internal historical-climatological database of the Institute of Geography, Masaryk University, Brno, with a particular focus on records from southwest Bohemia, further complemented by other local documentary evidence such as chronicles from the studied area and regular reports from the I. R. Patriotic Economic Society

of Bohemia between 1822–1847 CE (Resultate, 1828; Neue Schriften, 1830–1847; Verhandlungen und Mitteilungen, 1849, 1850). These records also contain annual information related to agriculture and grain yields for some selected domains or locations.

Climatic patterns in the region, analyzed for 1789–1849 CE, were based on long-term series reconstructed for three basic climatic variables for the Czech Lands territory since 1501 CE:

(i) temperature series of Central Europe, reconstructed from documentary-based temperature indices of Germany, Switzerland, and the Czech Lands (1500–1854 CE), and mean temperatures from 11 Central European stations (from 1760 CE onwards) by Dobrovolný et al. (2010), which is fully representative of the Czech territory (see Brázdil et al., 2022);

(ii) precipitation series of the Czech Lands, reconstructed from documentary-based precipitation indices (1501–1854 CE) and mean areal precipitation totals (from 1804 CE onwards) by Dobrovolný et al. (2015);

(iii) the self-calibrated Palmer Drought Severity Index (scPDSI) of the Czech Lands, calculated by Brázdil et al. (2016a) from temperature (Dobrovolný et al., 2010) and precipitation (Dobrovolný et al., 2015) reconstructions.

### 3 Methods

For the purposes of this study, the following four annual grain variables collected from domain granary records (Sect. 2.2) were selected:

**(i) Quantity of borrowed grain**

The quantity of borrowed grain for sowing in the current year primarily reflected a poor grain harvest or its failure in the preceding year. However, it could also include grain borrowed for new spring sowing when winter cereals had perished due to an unfavorable winter.





**(ii) Quantity of returned grain**

The quantity of returned grain indicated a good harvest in the current year.

**(iii) Quantity of grain stored in the granary at the end of the year**

The quantity of grain stored in the granary at the end of the current year represented a balance between the quantities of borrowed and returned grain. An increase in the annual grain quantity indicated a good harvest in the current year, and *vice versa*.

**(iv) Total debt of serfs at the end of the year**

The debt of serfs at the end of the current year reflected not only the balance between the quantities of borrowed and returned grain in that year but also the cumulative debt from preceding years.

The simplified relationships between the four variables (i)–(iv) are shown in Fig. 3. A poor harvest in year *n* forced serfs to borrow grain for sowing in year *n+1*. If the harvest in year *n+1* did not allow them to return the same quantity or more, it was

reflected in a reduced quantity of grain stored, which also contributed to an increased debt for the serfs. On the other hand, a good harvest in year *n+1* could lead to an increase in stored grain and a reduction in serf debt. Similarly, a good harvest in given year *n* would result in an increased quantity of returned grain in the same year, contributing to more grain stored in the granary and a decreased debt for the serfs.

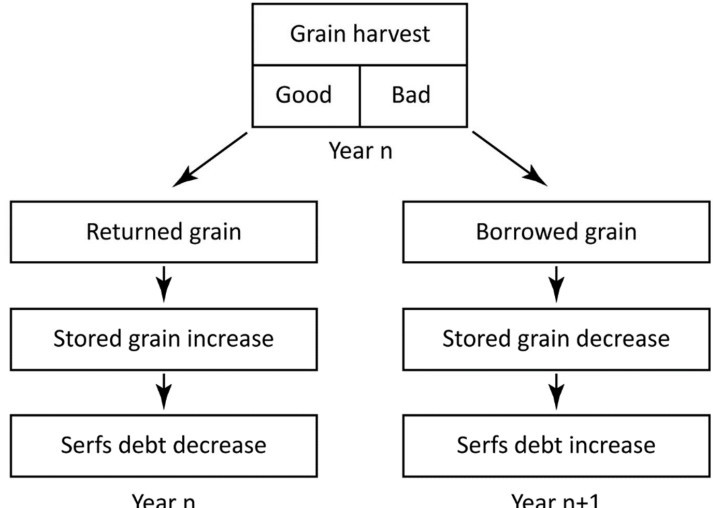

**Figure 3. Scheme of relationships between the four grain variables from the granary records.**

Only 15 out of a total of 17 considered domains, which had available data on grain variables (i)–(iv) for a sequence of years, were selected for further analyses. To select years of extreme harvests, the series of four variables for each corresponding domain were detrended using a high-pass filter. The overall arithmetic mean and standard deviation of these detrended series



were then used to select years with extreme grain harvests for each domain, applying thresholds defined by the mean plus
multiples of the standard deviation, ranging from 0.5 to 5.0 in 0.5 steps for the corresponding variables (i)–(iv). Since
individual domain records were not preserved for each year of the 1789–1849 CE period, the harvest extremeness of years
for all 15 domains together was described by involving different weighted grain indices. These indices were calculated as the
sum of the highest multiples of the standard deviation (from 0.5) achieved during a particular year across all domains,
divided by the number of domains that had any related (non-missing) data for that year. By applying this method to the four
grain variables, the following four types of weighted grain indices were obtained:

(i) weighted bad harvest index (WBHI),

(ii) weighted good harvest index (WGHI),

(iii) weighted stored grain index (WSGI: WSGI-, more grain borrowed than returned; WSGI+, more grain returned than
borrowed),

(iv) weighted serf debt index (WSDI: WSDI+, more grain borrowed than returned; WSDI-, more grain returned than
borrowed).

Combining the above indices allowed for the identification of years corresponding to bad or good harvests. These years were
then cross-referenced with available documentary evidence on weather and harvest conditions related to the Sušice region in
southwest Bohemia or other more distant areas in Bohemia. Three climatic variables during bad and good harvest years were
used as causal factors in a composite analysis (von Storch and Zwiers, 1999; Li and Dolman, 2023). This analysis was
applied to groups of selected bad and good harvest years, with the associated seasonal temperature, precipitation, and scPDSI
patterns for rye, barley, and oats presented in box plots (including median, upper and lower quartiles, maximum, and
minimum). To demonstrate statistically significant differences between the climate variables in bad and good harvest years,
t-test for means, the median test, and F-test for variances were applied (Higgins, 2004).

## 4 Results

### 4.1 Climatic patterns in 1789–1849

Temperature, precipitation, and drought (expressed by scPDSI) series for the Czech Lands during the 1789–1849 CE period
were used to describe the climatological background of grain production. These three climate variables were analyzed for all
seasons due to the impact of weather patterns on both winter and spring crops throughout the growing cycle, from sowing to
harvest. Their seasonal fluctuations, along with linear trends, are shown in Fig. 4. Linear trends were calculated using the
non-parametric Theil-Sen method, which is generally robust against outliers in time series (Sen, 1968; Theil, 1992). To
determine the presence of statistically significant linear trends, the non-parametric Mann-Kendall test was applied (Mann,
1945; Kendall, 1975). Despite substantial inter-annual variability in the seasonal climate series, the identified linear trends
were not uniform. Temperature series showed a cooling tendency (decreasing trends) for all seasons, but these trends were
statistically significant only for spring (MAM; –0.21 °C/10 years, $p<0.05$) and summer (JJA; –0.16 °C/10 years, $p<0.10$).



For precipitation totals, winter (DJF) and autumn (SON) showed small decreasing trends, while MAM and JJA showed increasing trends (with the MAM increase being significant: 4.9 mm/10 years, p<0.05). Despite these mixed trends, all seasons showed statistically significant increases in wetness, as expressed by the scPDSI, even with significance at the p<0.01 level. The fluctuations in these three climate variables indicated a shift from generally warmer and drier conditions to

cooler and wetter patterns around 1810 CE.

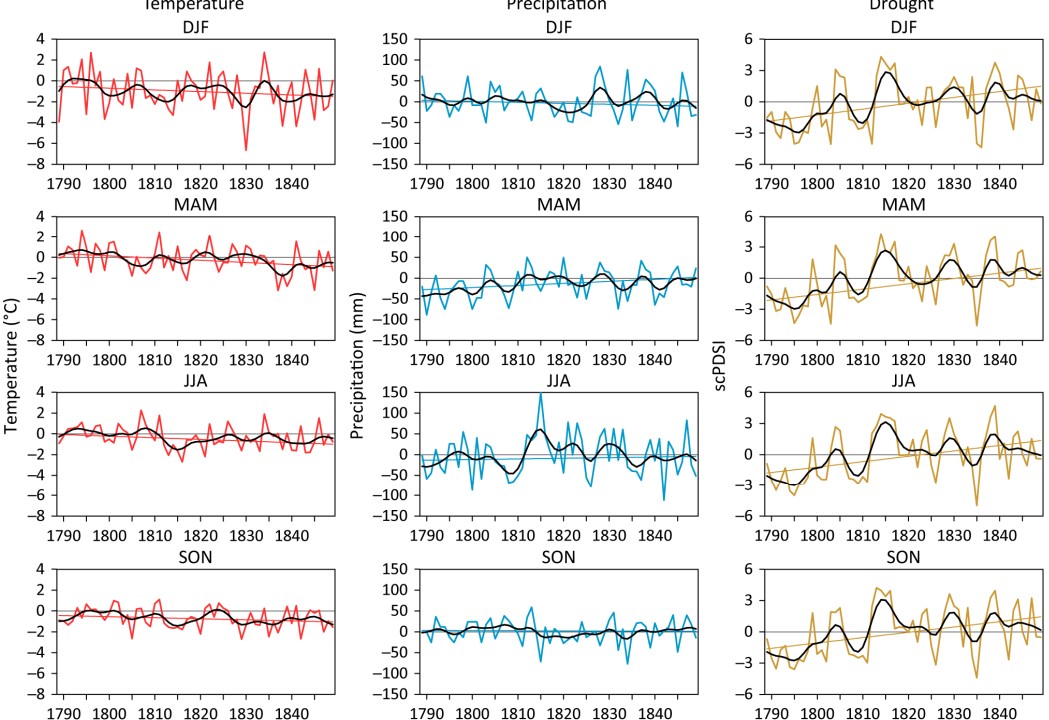

**Figure 4. Fluctuations and linear trends in mean winter (DJF), spring (MAM), summer (JJA), and autumn (SON) temperatures, precipitation totals, and scPDSI in the Czech Lands during the 1789–1849 CE period. The values are presented as deviations relative to the 1961–1990 reference period and smoothed using a 10-year Gaussian filter.**

**4.2 Grain harvest proxies in 1789–1849**

Because the quantities of borrowed, returned, and stored grain in granaries, along with the debt of serfs (see Sect. 3), for individual domains typically covered different numbers of years during the 1789–1849 period, Fig. 5 shows the annual number of domains for which related data were available. Of the 17 considered domains (see Table 1), no granary data were available for Dolejší Krušec and Lipová Lhota. The slow increase in the number of domains with available data reflects the





225 initial challenges in implementing the executive instructions for granary creation following the publication of the emperor's patent on 9 June 1788. This gradual increase in data availability varied in duration; for example, data on borrowed grain became consistently available for at least ten domains by 1809 CE (Fig. 5a), while debt owed by serfs had reached this threshold by 1810 CE (Fig. 5d). In contrast, data on returned grain remained scarce for up to five domains until 1823 CE (Fig. 5b). The best-covered period for both borrowed and returned grain occurred between 1832 and 1845 CE, with data

230 available for ten or more domains. Records of stored grain were well covered from 1825 to 1845 CE (Fig. 5c), and records of serf debt were consistently available between 1810 and 1848 CE (Fig. 5d). However, data for the final years of the period, 1846–1849 CE, became scarce again. The trends described primarily concern data on rye and oats, the two dominant cereals. Only fragmentary data were recorded for barley, with records from either none or fewer than five domains. Continuous data for five domains related to barley were available only for serf debt between 1819 and 1836 CE (Fig. 5d).



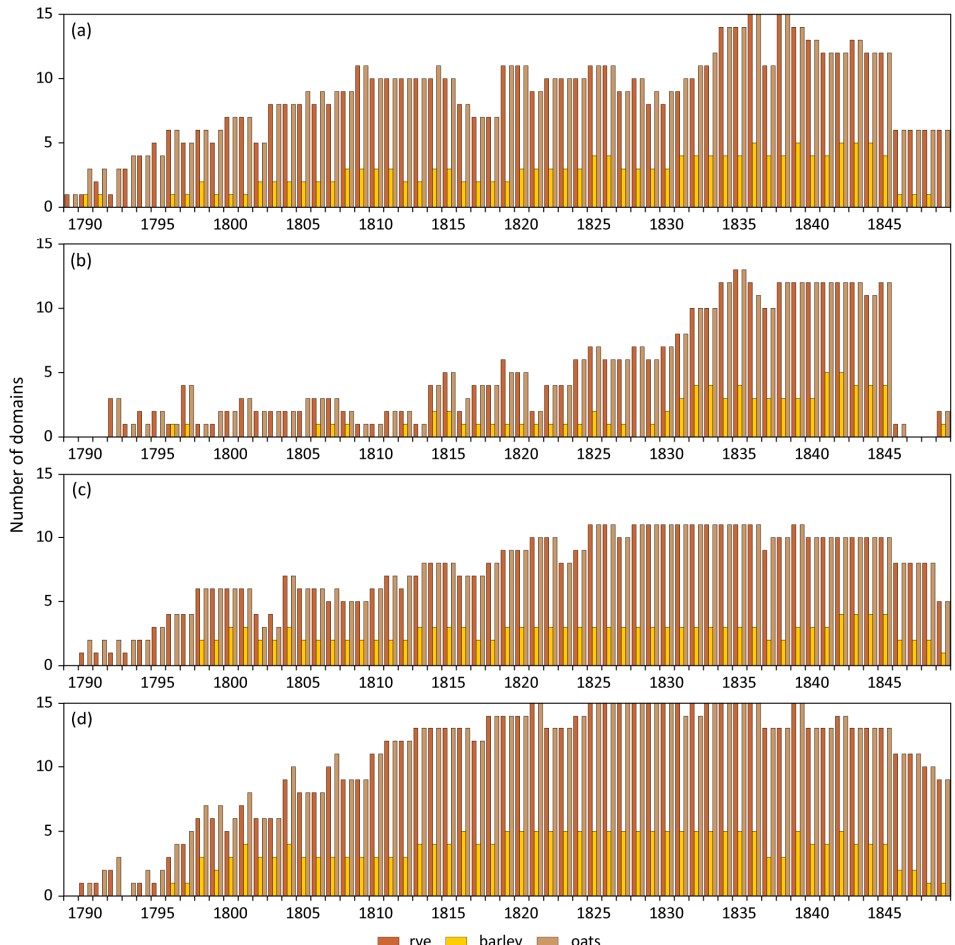


**Figure 5. Annual numbers of individual domains with available granary data, structured according to individual cereals (rye, barley, oats) in the Sušice region during the 1789–1849 CE period: (a) borrowed grain, (b) returned grain, (c) stored grain, and (d) debt of serfs.**

**4.3 Selection of extreme harvest years**

To select years of extreme grain harvests in the entire Sušice region during the 1789–1849 CE period, different annual weighted grain indices for each of the four selected grain variables, based on the methodology described in Sect. 3, were calculated. Bad/failed grain harvests were indicated by the weighted bad harvest index (WBHI), a decrease in the weighted stored grain index (WSGI-; more grain borrowed than returned), and an increase in the weighted serf debt index (WSDI+;





more grain borrowed than returned). Fig. 6 shows significant inter-annual variability in the three indices of bad harvests,
with varying degrees of overlap between their local peaks. Missing or zero values appeared particularly in the early years
following the establishment of public granaries, which is most evident for barley, represented by data from only six domains.
Table 2 presents, for all three indices (≥0.5), the individual years indicating poor grain harvests, along with the number of
domains for which data were available in the given year. While some extreme years were reflected in all three indices, others
were expressed in two indices (often for WSGI- and WSDI+), and many were indicated by only one index. Years identified
from a higher number of domains indicate greater regional representativeness of poor harvests compared to those derived
from a smaller number of domains, which may reflect more localized prominence.

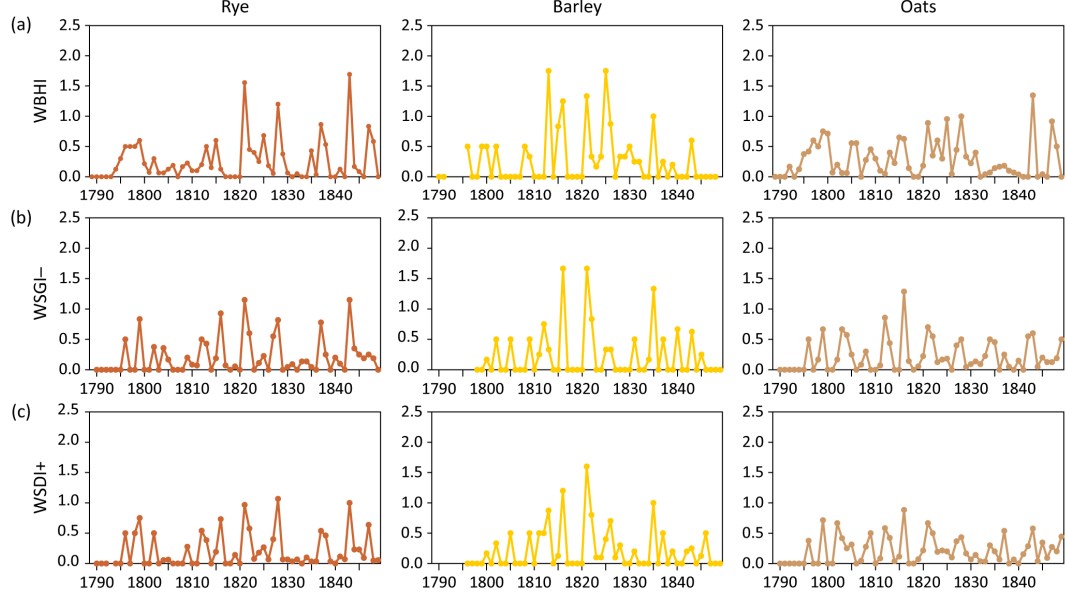

**Figure 6. Fluctuations in annual weighted grain indices in the Sušice region indicating bad harvests for rye, barley, and oats during the 1789–1849 CE period: (a) WBHI – weighted bad harvest index, (b) WSGI- – weighted stored grain index (decrease), (c) 255 WSDI+ – weighted serf debt index (increase).**

**Table 2. Bad grain harvest years ordered according to values of weighted grain indices ≥0.5 (≥1.0 in italics) derived from data from contribution granaries of 15 domains in the Sušice region during the 1789–1849 CE period: WBHI – weighted bad harvest index, WSGI- – weighted stored grain index (decrease), WSDI+ – weighted serf debt index (increase), No. – number of domains with data available. The real year of a bad grain harvest *n* is indicated by WBHI for the year *n + 1*.**

| Rye | | | | | | Barley | | | | | |
|---|---|---|---|---|---|---|---|---|---|---|---|
| WBHI | | WSGI- | | WSDI+ | | WBHI | | WSGI- | | WSDI+ | |
| Year | No. | Year | No. | Year | No. | Year | No. | Year | No. | Year | No. |





Climate of the Past Discussions — Open Access EGU

| | | | | | | | | | | | |
|---|---|---|---|---|---|---|---|---|---|---|---|
| *1843* | *13* | *1821* | *10* | *1828* | *15* | *1813* | *2* | *1816* | *3* | *1821* | *5* |
| *1821* | *9* | *1843* | *10* | *1843* | *13* | *1825* | *4* | *1821* | *3* | *1816* | *5* |
| *1828* | *10* | 1816 | 7 | 1821 | 15 | *1821* | *3* | *1835* | *3* | *1835* | *5* |
| 1837 | 11 | 1799 | 6 | 1799 | 6 | *1816* | *2* | 1822 | 3 | 1813 | 4 |
| 1847 | 6 | 1828 | 11 | 1816 | 13 | *1835* | *4* | 1812 | 2 | 1822 | 5 |
| 1825 | 11 | 1837 | 9 | 1847 | 11 | 1826 | 4 | 1840 | 3 | 1826 | 5 |
| 1799 | 5 | 1822 | 10 | 1822 | 13 | 1815 | 3 | 1843 | 4 | 1805 | 3 |
| 1815 | 10 | 1827 | 10 | 1812 | 12 | 1843 | 5 | 1802 | 2 | 1809 | 3 |
| 1848 | 6 | 1796 | 4 | 1837 | 13 | 1796 | 1 | 1805 | 2 | 1811 | 3 |
| 1838 | 15 | 1812 | 6 | 1796 | 3 | 1799 | 1 | 1809 | 2 | 1812 | 3 |
| 1796 | 6 | | | 1798 | 6 | 1800 | 1 | 1831 | 3 | 1837 | 3 |
| 1797 | 5 | | | 1802 | 6 | 1802 | 2 | 1837 | 2 | 1846 | 2 |
| 1798 | 6 | | | | | 1808 | 3 | | | | |
| 1813 | 10 | | | | | 1830 | 3 | | | | |


| Oats | | | | | |
|---|---|---|---|---|---|
| WBHI | | WSGI- | | WSDI+ | |
| Year | No. | Year | No. | Year | No. |
| *1843* | *13* | *1816* | *7* | 1816 | 13 |
| *1828* | *10* | 1812 | 7 | 1799 | 7 |
| 1825 | 11 | 1821 | 10 | 1802 | 6 |
| 1847 | 6 | 1799 | 6 | 1821 | 15 |
| 1821 | 9 | 1803 | 3 | 1812 | 12 |
| 1799 | 6 | 1843 | 10 | 1843 | 13 |
| 1800 | 7 | 1804 | 7 | 1837 | 13 |
| 1815 | 10 | 1822 | 10 | 1809 | 9 |
| 1816 | 8 | 1842 | 10 | 1822 | 13 |
| 1797 | 5 | 1796 | 4 | | |
| 1823 | 10 | 1828 | 11 | | |
| 1805 | 9 | 1834 | 11 | | |
| 1806 | 9 | 1849 | 5 | | |
| 1798 | 6 | 1816 | 7 | | |





| 1848 | 6 |
| 1843 | 13 |

The overview of bad grain harvest years in Table 2 can, in some cases, be complemented by short notes from granary records that report bad harvests in connection with the effects of adverse weather. For example, records from the Albrechtice domain for 1796 CE mentioned that "*oats were burned due to great heat*" (AS8). The following short remarks concerned
grain returns to the granary of the Hlavňovice domain: 1797 CE: "*No grain was returned due to weather damage.*"; 1799 CE: "*Nothing returned due to low yield and many in-kind batches.*"; 1801 CE: "*Nothing returned due to poor* [grain] *growth and bad harvest.*" (AS10). A similar report was noted in the Kolinec domain in 1797 CE (AS7): "*Serfs returned nothing this year due to bad weather.*" In 1802, serfs at the Velhartice domain were affected by strong hail (*Wetterschlag*) and returned no grain to the granary (AS21). The granary records from the Žichovice domain reported four instances of "*damage due to*
*weather*" – from hailstorms – in 1803, on 21 August 1806, on 16 August 1808, and in 1812 (AS17). A bad grain harvest was also mentioned in the granary notes of the Dlouhá Ves domain for 1828 CE (AS13): "*Bad harvest this year, serfs could not return grain borrowed in the spring, and after autumn sowing, they had only a minimum to eat; oats went bad due to wet weather.*" In 1844 CE, at the Albrechtice domain, "*the harvest was very bad, accompanied by wet weather, so that serfs were not able to cover their own needs from their yields and had to buy grain*" (AS8). A bad harvest also occurred in the
following year, 1845, when "*serfs had to buy grain at high prices to cover* [their] *grain debt*" (ibid.).

As for the years of a good grain harvest, they were indicated by the weighted good harvest index (WGHI), an increase in the weighted stored grain index (WSGI+; more grain returned than borrowed), and a decrease in the weighted serf debt index (WSDI-; more grain returned than borrowed) (Fig. 7, Table 3). The features described above for the three grain indices of bad harvests were generally valid for those indicating good harvests as well. However, notable differences appeared in the
WGHI compared to the WBHI for rye, with a long period of zero values from 1796 to 1807 CE and one dominant peak in 1823, which was also well expressed in WSGI+ and WSDI-. Similarly, the WGHI for barley suffered from only sporadic data until 1814 CE, with an outlier of 2.5 in 1808 for the Mačice domain.





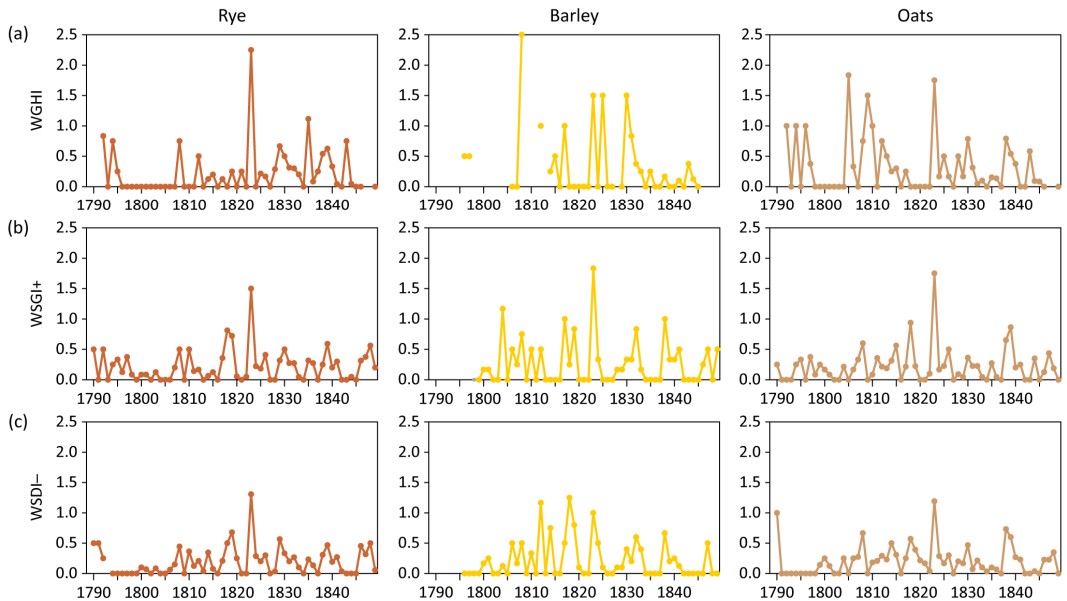

**Figure 7. Fluctuations in annual weighted grain indices indicating good harvests for rye, barley, and oats in the Sušice region**
**during the 1789–1849 CE period: (a) WGHI – weighted good harvest index, (b) WSGI+ – weighted stored grain index (increase),**
**(c) WSDI- – weighted serf debt index (decrease).**

**Table 3. Good grain harvest years ordered according to values of weighted grain indices ≥0.5 (≥1.0 in italics) derived from data**
**from contribution granaries of 15 domains in the Sušice region during the 1789–1849 CE period: WGHI – weighted good harvest**
**index, WSGI+ – weighted stored grain index (increase), WSDI- – weighted serf debt index (decrease), No. – number of domains**
**with data available.**

| | Rye | | | | | | Barley | | | | |
|---|---|---|---|---|---|---|---|---|---|---|---|
| WGHI | | WSGI+ | | WSDI- | | WGHI | | WSGI+ | | WSDI- | |
| Year | No. | Year | No. | Year | No. | Year | No. | Year | No. | Year | No. |
| *1823* | *4* | *1823* | *8* | *1823* | *13* | *1808* | *1* | *1823* | *3* | *1818* | *4* |
| *1835* | *13* | 1818 | 8 | 1819 | 14 | *1823* | *1* | *1804* | *3* | *1812* | *3* |
| 1792 | 3 | 1819 | 9 | 1829 | 15 | *1825* | *2* | *1817* | *2* | *1823* | *5* |
| 1794 | 2 | 1839 | 11 | 1790 | 1 | *1830* | *2* | *1838* | *2* | 1819 | 5 |
| 1808 | 2 | 1848 | 8 | 1791 | 1 | *1812* | *1* | 1819 | 3 | 1814 | 4 |
| 1843 | 12 | 1790 | 1 | 1818 | 14 | *1817* | *1* | 1832 | 3 | 1838 | 3 |
| 1829 | 6 | 1792 | 1 | 1848 | 10 | 1831 | 3 | 1808 | 2 | 1832 | 5 |
| 1839 | 12 | 1808 | 5 | | | 1796 | 1 | 1806 | 2 | 1806 | 3 |



| 1838 | 12 | 1810 | 6 | | 1797 | 1 | 1810 | 2 | 1808 | 3 |
|------|----|------|----|---|------|---|------|---|------|---|
| 1812 | 2 | 1830 | 11 | | 1815 | 2 | 1812 | 2 | 1817 | 4 |
| 1830 | 7 | | | | | | 1841 | 3 | 1824 | 5 |
| | | | | | | | 1847 | 2 | 1847 | 2 |
| | | | | | | | 1849 | 1 | | |

| Oats | | | | | |
|------|------|------|------|------|------|
| WGHI | | WSGI+ | | WSDI- | |
| Year | No. | Year | No. | Year | No. |
| *1805* | *3* | *1823* | *8* | *1823* | *13* |
| *1823* | *4* | 1818 | 8 | *1790* | *1* |
| *1809* | *1* | 1839 | 11 | 1838 | 13 |
| *1792* | *3* | 1838 | 10 | 1808 | 9 |
| *1794* | *1* | 1808 | 5 | 1839 | 15 |
| *1796* | *1* | 1815 | 8 | 1818 | 14 |
| *1810* | *1* | 1826 | 11 | 1814 | 13 |
| 1838 | 12 | | | | |
| 1830 | 7 | | | | |
| 1808 | 2 | | | | |
| 1812 | 2 | | | | |
| 1843 | 12 | | | | |
| 1839 | 12 | | | | |
| 1813 | 1 | | | | |
| 1825 | 7 | | | | |
| 1828 | 7 | | | | |

### 4.4 Extreme harvest years, weather and climate

### 4.4.1 Extreme harvest years and documentary data

Ten years of extremely bad and seven years of extremely good harvests of rye, barley, and oats (or at least two of them) were selected for comparison with documentary data on weather and related phenomena from the Sušice region and its





surroundings. This analysis combines information from the weighted grain indices shown in Figs. 6 and 7 and Tables 2 and 3, along with their logical evaluation and remarks in granary records. Due to the substantial amount of documentary data, we present here only a representative selection of corresponding records that confirm the character of especially extreme years
(for location of further cited places see Fig. A1).

**(A) Bad grain harvest**

Selected years of poor grain harvests were characterized by different reports from documentary sources as follows:

**(i) 1795**: The reeve in Milčice (central Bohemia), František Jan Vavák, mentioned frequent rainy days in summer, leading many farmers to harvest grain that was moist and wet. The harvest was better for rye than in the preceding year, 1794, and
good for barley and oats, but poor for wheat (Skopec, 1916). The rainy weather in summer was confirmed by records from Bělá nad Radbuzou, where on 24–26 July parishioners prayed for a nice weather (AS25). Poor winter cereals and good spring cereals were harvested in Klášter Hradiště nad Jizerou (Šimon, 1927).

**(ii) 1798**: A note in the granary records of the Kolinec domain for the year 1798 mentioned "*Accounting of tax reduction for damage caused by weather*" (AS7). A source from Skuhrov nad Bělou reported a very cold and rainy spring and summer,
with only about two weeks of warm and sunny weather. In higher elevations, much of the grain went bad (Robek, 1976). Reports from Milčice indicated less straw from rye and barley but good grain quality, along with a lower-than-usual oat harvest at higher elevations (Skopec, 1918). A scanty harvest and rising grain prices during the harvest were reported in Noviny pod Ralskem (AS24).

**(iii) 1811**: Dry conditions from May to June and warm weather from May to July in the Czech Lands contributed to an
earlier onset of the grain harvest, which was very poor, as documented by many sources throughout the country (see Brázdil and Trnka, 2015). While the granary records of the Velhartice domain noted only "*bad harvest*" for 1811 without specifying the cause (AS21), other sources from central Bohemia linked the bad harvest to dryness. For instance, in Dřínov, cereals dried out immediately after blossoming, and in Hostín, only a small amount of rye was reported (Robek, 1974). The parson in Žitenice, František Jindřich Jakub Kreybich, reported very sparse rye and wheat, short in stature, along with spring cereals
(AS3). František Jan Vavák noted in his records from Milčice that "*rye and wheat are short, barley sparse, and everything of a small ear*" (Jonášová, 2009, p. 93).

**(iv) 1812**: Potential grain yields in 1812 were reduced by hail damage, as reported in seven villages of the Žichovice domain (AS17). Similarly, hail damage was noted in the Nalžovy domain (AS6): "*Part of* [the annual] *debt* [of serfs] *was remitted to villages damaged by* '*Wetterschlag*' [hailstorm]." František Jan Vavák recorded for Milčice that rye lay on the ground,
overgrown with grass, due to frequent rains, and that some fields were infested with beetles eating the grain (Jonášová, 2009). A poorer grain harvest was also reported in Ouholice (Neruda, 1862). In Budenice, the grain harvest was prolonged due to cold weather until September (Komárek, 1911), while in Hora Svatého Šebestiána, adverse weather prevented grain harvesting altogether, leaving no grain for the next sowing (Binterová, 2000).

**(v) 1815**: Georg Mayer in Fleky reported a cold and wet year with a poor grain harvest and a large number of mice (Blau,
1908). Josef Schück, in his records from Litoměřice, mentioned a very scant harvest across all of Bohemia due to incessant





wet weather (Bachmann, 1911). František Jan Vavák from Milčice noted a long period of frost, followed by dry weather, which was then replaced by continuous rain. This caused many rye ears to lack grain, barley to have short ears and small size, and overall, *the grain harvest was woeful, and peasants carried wet grain to barns, making threshing difficult*" (Jonášová, 2009, p. 388). The harvest failure of 1815 coincided with the large Tambora volcanic eruption in April, and the

following year, 1816, became known as "the year without a summer" (Luterbacher and Pfister, 2015; Brázdil et al., 2016b).

**(vi) 1820**: The school chronicle of Bukovník mentioned a severe thunderstorm with torrential rain and hail on 23 May, that damaged heavily fields and were followed by great drought (AS29). Very dry summer was reported also for Bělá nad Radbuzou (AS25).

**(vii) 1821**: The poor grain harvest of 1821 was reflected in the granary remarks of the Dlouhá Ves domain (AS13): 1821:

"*Half* [the grain] *went bad due to weather.*" 1822: "*Serfs returned only a small amount of grain due to the poor harvest of the previous year* [1821] *and the not much better harvest of this year* [1822]." Augustin Kalach from Obora mentioned a late harvest, partly due to unripe grain and partly due to incessant rains (Robek, 1979). During the rainy August, a procession beseeching favorable weather for the grain harvest was organized on 19 August in Jablonné v Podještědí (AS24). The memories of Jan Chlebeček from Chrudim cited a cold, wet, and rainy year, with rains during the August harvest that

negatively impacted rye (AS23).

**(viii) 1827**: The school chronicle of Bukovník reported a "*bad harvest due to great drought*" (AS29). Records from Řenče mentioned delayed spring sowing (end of April to early May) due to excessive wetness and generally average yields, although "*barley was bad everywhere*" (Urban, 1999, pp. 45–46). A great drought also contributed to a poor grain harvest in Hostín (Robek, 1974). The harvest in Hřivice was poor in grain, leading to rising prices by August (Robek, 1979). According

to a report by the I. R. Patriotic Economic Society of Bohemia, grain yields were negatively influenced by a rainy March and April, which delayed spring sowing, followed by rainy weather in June during rye and wheat blossoming, and then very hot and dry weather in the subsequent two months (Neue Schriften, 1830).

**(ix) 1836**: The chronicle from Lhovice reported a dry year with less wheat, average rye (more straw and less grain), as well as average barley and oats (AS27). The municipal chronicle of Poleň mentioned a cold spring lasting until May, followed by

warmer weather with some rain, and then a summer drought, which resulted in rye with good straw but weak grain, average wheat, and very poor barley and oats (AS26). In neighboring Poleňka, the wheat harvest was also very poor (AS28). At Hřivice, grain dried up, with very bad rye and slightly better barley (Robek, 1979). A report from the I. R. Patriotic Economic Society of Bohemia described this year as generally characterized by higher temperatures and lighter rains, especially in summer; for example, in Litoměřice, rye was sparse, wheat short, and barley and oats varied in growth (Neue

Schriften, 1838).

**(x) 1842**: The very bad harvest of 1842 in the Czech Lands was related to the extremely dry conditions from April to September, as documented by many sources (see Brázdil and Trnka, 2015). This drought extended beyond the Czech Lands to a broader European region (Brázdil et al., 2019a). The granary records of the Albrechtice domain mentioned "*a bad harvest* [Missernte] *in the region*" in this year (AS8). A severe drought with almost no rain in Lhovice after spring sowing





led to a very scant harvest, with nearly no oats (AS27). Similarly, in Budětice, the absence of rain since spring caused poor cereal growth and a meager harvest, with no oats at all (AS30). The chronicle of Bukovník also reported particularly poor barley and oats due to the severe drought (AS29), while in Poleň, rye was average (but very short), with bad harvests for the other three cereals (AS26).

**(B) Good grain harvest**

The selected years of good grain harvests were characterized by different reports from documentary sources as follows:

**(i) 1808:** Although Czech meteorological data indicated dry conditions in March, May, and July–August (Brázdil and Trnka, 2015), the grain harvest was affected in different ways. František Jan Vavák in Milčice mentioned that the dry and warm weather caused fast ripening of cereals, resulting in their drying out with only a little grain (Skopec, 1938). However, František Jindřich Jakub Kreybich in Žitenice described the year as good and fruitful for cereals, with good harvests of

winter crops like rye and wheat, although spring crops like barley fared worse (AS1). A very good harvest was noted in Dřínov in central Bohemia (Robek, 1974).

**(ii) 1810:** František Jindřich Jakub Kreybich characterized the harvest in lowlands as average, but in higher altitudes ("mountains"), both winter and spring crops, especially rye and oats, were very good (AS2). An average grain harvest was also reported for Dřínov (Robek, 1974).

**(iii) 1819:** The parish chronicle from Srní reported a plentiful year with cheaper rye (AS4). The school chronicle of Bukovník cited "*a great cheapness in grain*" (AS29). A plentiful year was also reported in the chronicles from Bělá nad Radbuzou (AS25) or Zhůří (AS32).

**(iv) 1823:** A good grain harvest was reported by Georg Mayer in Fleky (Blau, 1908). Václav Jan Mašek from Řenče noted that 1823 brought "[…] *nice times and also fertile rain. From it followed God's blessing on fields in the harvest during the*

*whole summer, but God sent us a scourge of hail in mid-harvest* [8 August]. *Not too much damage was done, only to some oats* […]" (Urban, 1999, p. 42). Records from Jablonné v Podještědí cited an abundant grain harvest due to favorable weather (AS24). The agricultural summary of the I. R. Patriotic Economic Society of Bohemia for 1823 mentioned a good autumn sowing (in 1822), followed by a severe winter that delayed spring sowing until late April in mountainous regions, resulting in a very good grain harvest by the second part of August (Resultate, 1828).

**(v) 1830:** An average harvest was reported in the chronicle from Bukovník (AS29) similarly as in Srní, where the fertility of this year was evaluated generally as average (AS4). The parish chronicle from Železná Ruda reported high summer temperatures leading to thunderstorms, but they did not occur in the surroundings and grain harvest was identified as good (AS33). According to reports from the I. R. Patriotic Economic Society of Bohemia, the rye harvest turned out to be better than expected after poor autumn sowing (1829) and a very severe winter 1829/1830; yields of barley and oats were evaluated

as good (Neue Schriften, 1833).

**(vi) 1838:** The local chronicle of Lhovice reported "*such an abundant harvest as not remembered for many years, both in grain and straw*" (AS27). An abundant harvest was also noted in Hojsova Stráž (AS31). The parish chronicle of Kašperské Hory cited average rye but excellent oats (AS5). A good harvest of all four cereals was reported in the chronicle from Poleň





(AS26), and similarly in Poleňka, where the chronicler added "*such a good harvest* [of grain] *had not been recorded for*
*many years*" (AS28). The I. R. Patriotic Economic Society summarized reports of harvests from 12 different locations in
Bohemia as follows: wheat – good for both grain and straw; rye – rather average; barley – partly very good, partly below
average; oats – good, particularly for grain (Neue Schriften, 1840).

**(vii) 1839:** According to records from Hojsova Stráž, severe winter conditions affected meadows and fields, but favorable
spring and summer weather contributed to a good grain harvest (AS31). A source from Lhovice cited a good harvest with
abundant straw but less grain (AS27). The chronicle from Poleň reported average yields for rye, wheat, and barley, but a
good harvest of oats (AS26). Information on grain harvests from 17 different locations in Bohemia was summarized in the
report of the I. R. Patriotic Economic Society as follows: wheat – good for both grain and straw; rye – regionally variable,
from very good to poor; barley – good, better for grain than straw; oats – good (Neue Schriften, 1841).

### 4.4.2 Climate and extreme harvest years

Table 4 lists the bad and good harvest years separately for rye, barley, and oats in the Sušice region, derived from the
calculated weighted grain indices in Tables 2 and 3. The number of bad harvest years was higher than good harvest years for
rye (15 vs. 13) and oats (11 vs. 7), while for barley, both datasets were represented by 12 years. These selected years of bad
and good harvests were further used in composite analyses to examine their relationships to temperature, precipitation, and
drought (scPDSI) patterns, represented by seasonal box plots of related climate variables (Fig. 8). Climatic patterns from the
previous SON to the JJA of the harvest year were considered for rye, while for barley and oats, the analysis was limited to
the period from DJF to JJA. For rye, years of bad harvests compared to good harvests were characterized by relatively higher
temperatures and lower precipitation totals from SON to JJA. This was particularly well expressed in the scPDSI deviations,
which showed much drier patterns for years of bad harvests. However, the high variation range in JJA precipitation totals
suggests that bad harvests could be influenced by either extremely high or extremely low precipitation totals. In contrast,
spring crops like barley and oats showed more ambiguous relationships. After a colder DJF in good harvest years, MAM
temperatures were higher in bad harvest years, similar to JJA for barley. For oats, JJA temperatures were comparable
between the two datasets. MAM and JJA precipitation totals were lower in years of bad barley harvests, while for oats, this
was only observed for MAM. Regarding the scPDSI, drier conditions characterized bad barley harvests from DJF to JJA,
while for oats, this pattern was observed only in JJA. Interestingly, lower median values for DJF and MAM scPDSI in years
of good oat harvests compared to bad harvests were accompanied by larger interquartile ranges, favoring wetter conditions.
A question remains as to what extent this differing relationship for oats results from the limited sample size of only seven
good harvest years compared to 11 bad harvest years. Testing the statistical significance of the differences in climate
variables between years of bad and good harvests showed significant results in only a few cases. Significant differences were
found in the means of MAM temperatures ($p<0.10$ for barley and $p<0.05$ for oats) and SON scPDSI for rye ($p<0.10$), in the
median for MAM temperatures for oats ($p<0.05$), and in variability for DJF temperatures ($p<0.10$ for rye and $p<0.05$ for
oats) and JJA temperatures ($p<0.05$ for barley).





**Table 4. Years of bad and good harvests of rye, barley, and oats in the Sušice region during the 1789–1849 CE period.**

| Cereal | Year of bad harvest |
| --- | --- |
| Rye | 1795, 1797, 1798, 1802, 1811, 1812, 1815, 1820, 1821, 1826, 1827, 1836, 1837, 1842, 1846 |
| Barley | 1801, 1804, 1811, 1812, 1815, 1820, 1821, 1825, 1827, 1834, 1836, 1842 |
| Oats | 1795, 1798, 1802, 1811, 1812, 1815, 1820, 1821, 1827, 1842, 1848 |

| Cereal | Year of good harvest |
| --- | --- |
| Rye | 1790, 1792, 1808, 1810, 1818, 1819, 1823, 1829, 1830, 1838, 1839, 1847, 1848 |
| Barley | 1806, 1808, 1810, 1817, 1819, 1823, 1824, 1830, 1832, 1833, 1838, 1847 |
| Oats | 1807, 1808, 1823, 1826, 1830, 1838, 1839 |

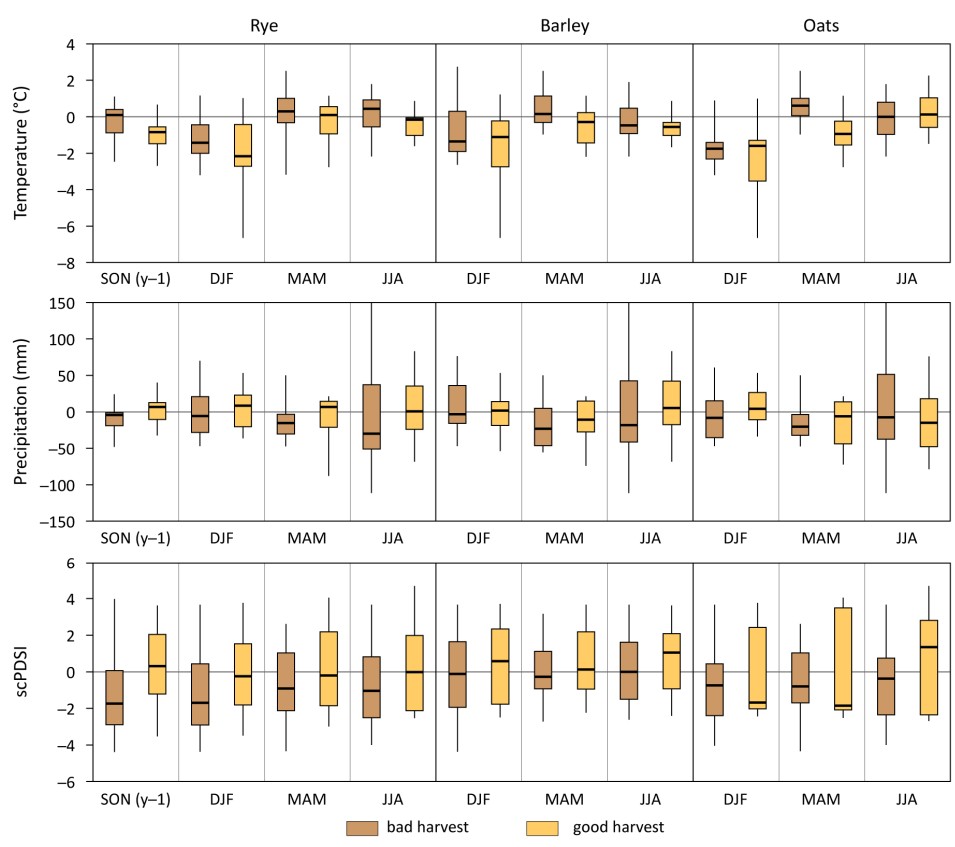






**Figure 8. Composite analysis represented by boxplots (median, upper and lower quartiles, maximum, and minimum) of seasonal (DJF – winter, MAM – spring, JJA – summer, SON (y–1) – autumn of the preceding year) mean temperatures, precipitation totals, and scPDSI for the Czech Lands during years of bad and good harvests of rye, barley, and oats in the Sušice region from 1789–1849 CE. Climate variables are expressed as deviations relative to the 1961–1990 reference period.**

## 5 Discussion

### 5.1 Data uncertainties

Various types of uncertainties are present in the granary data due to the multiple and sometimes competing purposes and interests associated with them, as well as their connection to a dynamic agricultural ecology. These factors contribute to some data gaps during the period analyzed. For instance, when no grain was available in the granary, it was impossible to borrow grain, and no grain circulation would be recorded, even in the case of a severe grain failure. For example, in 1799, no grain was returned to the granary at the Velhartice domain due to a bad harvest (AS21). Similarly, in the two subsequent years, serfs did not return any grain to the granary, as it was supplied to armies instead (ibid.). A comparable situation occurred in 1802, when the grain was damaged by a severe hailstorm (ibid.). At the Mačice domain in 1813, serfs used all available grain for "*army depots, transports, fresh draught horses* [Vorspann], *and faraway cartloads* [weite Fuhren]," leaving nothing to be delivered as debt payment (AS15).

A sudden decrease in grain in the granary could result not only from loans but also from grain sales, authorized by the corresponding officers. For example, a sale of 80.5 *měřice* (4,950 L) of rye and 27.3 *měřice* (1,679 L) of oats was reported at the Kněžice domain in 1838 (AS12), and 150 *měřice* (9,224 L) of rye from the Dlouhá Ves domain in 1839 (AS13). At the Nalžovy domain, starting from 1793 CE, nearly every following year, 34 *měřice* (2,091 L) of oats, and from 1839 CE, 52 *měřice* (3,198 L) of oats were given to the surgeon (AS6). In 1843 and 1844, 42 *měřice* (2,583 L) of rye were provided to serfs affected by fire (ibid.). At the Kašperské Hory domain, 479 *měřice* (29,454 L) of rye from the granary was distributed interest-free in 1837 CE among serfs for three years due to the danger of worms in the granary (AS9). Loss of grain in the granary could also have been related to thefts, as was the case in the Hrádek domain in 1812 (8 měřice of rye and 5 *měřice* of oats, i.e. 492 L and 307 L, respectively) or in 1828 (58 *měřice* of oats, i.e. 3,566 L) (AS11). In 1817 CE, at the Žikov domain, the regional office decided that the preceding amount of grain was not sufficient for sowing the serfs' fields and requested 436 *měřice* 5 1/16 *měřička* (26,829 L) of rye and 523 *měřice* 10 5/16 *měřička* (32,199 L) of oats from the serfs (AS22). Due to the lack of a public granary building, in 1827 CE all the grain in the granary was divided among the serfs, and their debt was decreased accordingly, while in the following years, one half always had to be delivered back (ibid.).

The selling of grain from granaries was considered in the analysis of the quantity of grain stored in the granary at the end of the year. Before the calculation of WSGI, the quantity of grain sold in year $n$ was added back to this year for comparison with the quantity of the preceding year, $n – 1$. However, it was not considered in comparing the grain quantities in year n with the following year, $n + 1$.



Some domains experienced territorial changes throughout the entire 1789–1849 CE period due to the sale of small parts of their land. Nevertheless, these had only a minor impact on both the number of active peasants and the extent of arable land.

The results of these changes were unsubstantial, affecting only a few peasants, and the incorporation of new areas often failed to influence the original accounting if crop fund administrations already existed. For example, Kolinec, which became part of the Nalžovy domain in 1837, remained a self-contained entity afterward. The same occurred in Dolejší Těšov, which joined the Kundratice domain in 1802. Sometimes, the borrowing of grain from crop funds in justified cases remained open to serfs from other domains, even when the borders of a particular domain were not changed.

The integrity of collected grain data was also partly influenced by issues already criticized by contemporary state administration, such as the complete lending out of the crop fund and the procrastination of repayments, which could last several years. This meant that grain existed only in the accounts, and in crisis years, it was not possible to use the funds for their intended purpose, as was the case at the Žikov domain (AS22). To avoid distorting the data, it was necessary to account for all individual corrections in grain accounting (such as remission of interest), involving in-kind salaries for patrimonial

medical and clerical staff, or reductions in stored grain due to pests (e.g., mice) or official misappropriation.

Regarding the weather/climate datasets used, some biases for their use in this study must be noted. The studied region is in the marginal part of Bohemia, at middle and higher altitudes, which may result in different weather and climate effects on grain harvests compared to lowland or generally lower-altitude areas. This may apply to documentary reports from more distant locations from the Sušice region, where the impacts of colder and wetter patterns on grain harvests could differ.

Moreover, the reconstructions of temperature, precipitation, and scPDSI represent average patterns rather than specific values for the entire territory of the Czech Lands, which may vary to some extent in the Sušice region. This underscores the importance of using documentary data directly connected to this region or its nearby surroundings.

## 5.2 Broader context

European interest in maintaining public granaries reached its peak at the end of the 18th century. Repeated harvest failures,

increased demographic pressure and new economic ideas converged in a veritable granary-mania. Journals all over the continent debated their potential and drawbacks, expanding from modest establishments geared at stocking seed grain or army supplies, to major institutions able to balance market prices and feeding whole populations (Collet, 2019). Some academies even hosted European competitions for the best papers on the pitfalls and potentials of granaries (Reimarus, 1772). A minority of discussants saw them as an expensive challenge to free trade that would de-incentivise private

merchants. Many more envisioned them as a vital infrastructure against increasingly frequent weather shocks, a tool to curb profiteering and, crucially, as a tool to strengthen administrative control of the grain market, a key sector of pre-industrial economies. Their form, organisation and dispersion, however, responded to local constellations and needs (see Supplement for Prussia and Northern Europe).

Concerning the Czech Lands, different grain data from public granaries of 15 domains in the Sušice region in southwest

Bohemia during the 1789–1849 CE period allowed for the calculation of six different weighted grain indices, which were





then combined to identify years of bad or good harvests of rye, barley, and oats (see Sect. 3). To assess the explanatory ability of these indices in representing the same harvest characteristics, Spearman rank correlation coefficients were calculated between the indices used for delimiting bad and good grain harvests (Table 5). All correlation coefficients were statistically significant, at least at the $p<0.10$ level, and were generally higher between bad harvest indices than between

good harvest indices. Although the lowest correlations were observed for oats, those for barley were higher than for rye in the case of bad harvests, while the opposite was true for good harvests (except for correlation E – F between WSGI+ and WSDI-). As expected, the strongest correlation appeared between stored grain and serf debts (B – C and E – F), as demonstrated by the highest correlations.

**Table 5. Spearman rank correlation coefficients between series of weighted grain indices used for the delimitation of bad and good harvest years for rye, barley, and oats in the Sušice region during the 1789–1849 CE period. Statistical significance of coefficients: * $p<0.10$, ** $p<0.05$, *** $p<0.01$. Abbreviations of weighted grain indices: A – WBHI, B – WSGI-, C – WSDI+, D – WGHI, E – WSGI+, F – WSDI-.**

| Cereal | Bad harvest | | | Good harvest | | |
|--------|-------------|--------|--------|--------------|--------|--------|
|        | A – B       | A – C  | B – C  | D – E        | D – F  | E – F  |
| rye    | 0.51***     | 0.60***| 0.78***| 0.51***      | 0.45***| 0.74***|
| barley | 0.58***     | 0.62***| 0.80***| 0.45***      | 0.37** | 0.82***|
| oats   | 0.23*       | 0.34***| 0.78***| 0.26**       | 0.23*  | 0.71***|

The granary records were more representative in expressing bad harvests than good harvests (see Tables 2 and 3), which was

reflected in the higher number of years with weighted grain indices ≥0.5 (36 to 28 for rye, 38 to 35 for barley, and 37 to 30 for oats). Moreover, high proportions of indices for good harvests compared to bad harvests were derived from only one or two domains (25.0 % to none for rye, 51.4 % to 31.6 % for barley, and 26.7 % to none for oats). The presented results can be explained by a greater need for serfs to borrow grain from granaries rather than an obligation to return it. However, it needs to be acknowledged that barley in our region was cultivated less than rye and oats, which were considered the main cereals.

The final selection of extreme years identified 10 years with bad harvests and seven years with good harvests for at least two of the three cereals considered, well confirmed by documentary data from the analyzed region and other locations in Bohemia (see Sect. 4.4.1 – A, B). However, considering grain data availability (see Fig. 5), their spatial coverage by a variable number of domains (Tables 2 and 3), and the uncertainty in granary data (see Sect. 5.1), some extreme years with both bad and good grain harvests could remain unnoticed, as they are not well captured in the available granary records. A

potential spatiotemporal incompleteness of such man-made data is a typical feature of documentary data in historical climatological research (e.g., Brázdil et al., 2005; White et al., 2018b; Pfister and Wanner, 2021). When years of bad and good harvests were studied separately for each cereal, higher temperatures, lower precipitation totals, and intense droughts expressed by the scPDSI from SON of the preceding year to JJA of the given year characterized bad rye harvests. The same applied to MAM and JJA variables in the case of barley harvests. Less unambiguous patterns, as for barley, characterize

MAM and JJA climate variables in the case of oats (see Sect. 4.4.2). Statistically significant differences in mean seasonal



temperatures between years of bad and good harvests were found for MAM (barley, oats) and in temperature variability for DJF (rye, oats) and JJA (barley). However, climate–harvest yield associations can be regionally variable, as documented by Ljungqvist et al. (2023), who identified positive significant signals in JJA soil moisture for Sweden and warmer and drier DJF patterns for Switzerland, but negative signals in MAM and annual mean temperatures for Spain. Rácz (2023) attributed

poor winter wheat harvests in the Carpathian Basin during the first half of the 19th century to cold and wet October, cold March, dry May, dry-hot or wet June, and wet July and August months. The significance of MAM and JJA temperatures identified in southwest-Bohemian grain harvests coincides well, for example, with their impacts on grain prices in the Burgundian Low Countries during the 15th century (Camenisch, 2015) and on 56 analyzed grain price series in early modern Europe (Ljungqvist et al., 2022).

In the evaluation of general patterns during the 1789–1849 CE period, other important non-climatic factors must also be mentioned, particularly the time of the Napoleonic Wars in Europe between 1803 and 1815 CE. These wars were represented by a series of conflicts in which Great Britain, Russia, Prussia, and the Austrian Empire were the principal adversaries, with battles often occurring on or near their territories. The Czech Kingdom, as part of the Austrian Empire, significantly supported the Austrian and, to some extent, the coalition armies. French troops moved through Czech territory in 1805,

1809, and 1813 (e.g., Šedivý, 2001; Gates, 2003; Esdaile, 2019). Some response to military conflict is apparent in the granary records of the Hrádek domain, where in 1813, 410 *měřice* (25,211 L) of oats were distributed among serfs due to the danger of war (AS11). Yet, despite some oats being sold to the army as early as 1790–1792 CE in the Nalžovy domain (AS6), other information indicating support for armies in the study region is missing. This is notably different from Prussia, where military interests were instrumental in establishing public granaries (*cf.* Collet, 2010). The increasing importance of

potato growing was another factor influencing cereal production in the target area during the period analyzed. Potatoes, which found broad use in the less fertile foothills of the Šumava Mountains, disrupted the monopoly of cereals because they could fulfill the role of the main foodstuff instead.

Despite several studies addressing different historical, socio-economic, and political aspects of grain granaries in Europe (e.g., Franko, 1907; Černý, 1932; Kaplan, 1977; Teerijoki, 1993; Berg, 2007; Collet, 2010; Seppel, 2019; Løvdal, 2020), our

paper is the first to attempt using granary data for historical-climatological research. Specifically, it identifies bad or good grain harvests in relation to weather and climate patterns, situating them within the broader context of the Czech Lands in the late 18th and the first half of the 19th century. Additionally, it presents a methodology that can be used to analyze the large number of extant granary accounts available throughout Europe, which share the same or similar structures of record-keeping.

Although grain data mining from archival documentation of individual domain granaries is an extremely time-consuming process, the potential of such data is very high, especially if the data is combined comparatively across wider areas. It can also be used to distinguish the effects of broader weather/climate anomalies from more local or regional meteorological effects, such as frosts in specific geographic locations, downpours with subsequent field inundation, or hailstorms. The case study in this paper demonstrates how granary data can serve as proxies for grain harvest in relation to weather/climate



patterns. This could motivate further researchers to examine and potentially use granaries in other European countries as proxies for similar historical-climatological studies.

**6 Conclusion**

The analysis of grain records from public granaries of 17 individual domains in southwest Bohemia during the 1789–1849 CE period can be summarized as follows:

(i) Data recorded by public granaries documenting the quantity of borrowed seed grain, deposits after harvest, overall storage quantity, and the level of serfs' debt represent important proxy indicators of grain production, allowing the selection of years with bad and good yields.

(ii) The use of these four basic types of grain data enabled the calculation of six different weighted grain harvest indices, allowing for the identification of years with bad and good grain harvests. Despite incomplete granary data and their

relatively high uncertainty, the new methodology contributed to obtaining a sufficiently robust selection of years with bad and good harvests.

(iii) Ten selected years of extremely bad harvests and seven years of extremely good harvests, involving at least two of the three cereals considered (rye, barley, oats), were well confirmed and complemented by other documentary data on weather and related phenomena from the Czech Lands.

(iv) The number of identified years with bad harvests was higher than those with good harvests. Higher temperatures, lower precipitation totals, and intense droughts from SON to JJA distinguished bad rye harvests from good ones. A similar correlation characterized MAM and JJA patterns for barley, while MAM and JJA patterns for oats were less clear. Differences in climate variables between years of bad and good harvests were predominantly statistically non-significant.

(v) Grain records from public granaries may be used as important proxies for evaluating grain harvests in relation to weather

and climatic patterns. Extending the territorial scope of the analysis, together with the new methodology, opens up new research directions and offers great data and research potential for historical climatology. Given the ubiquity of such records in the 18th and 19th centuries, the approach developed here is applicable to a range of similar cases throughout Europe.



**Appendix A**

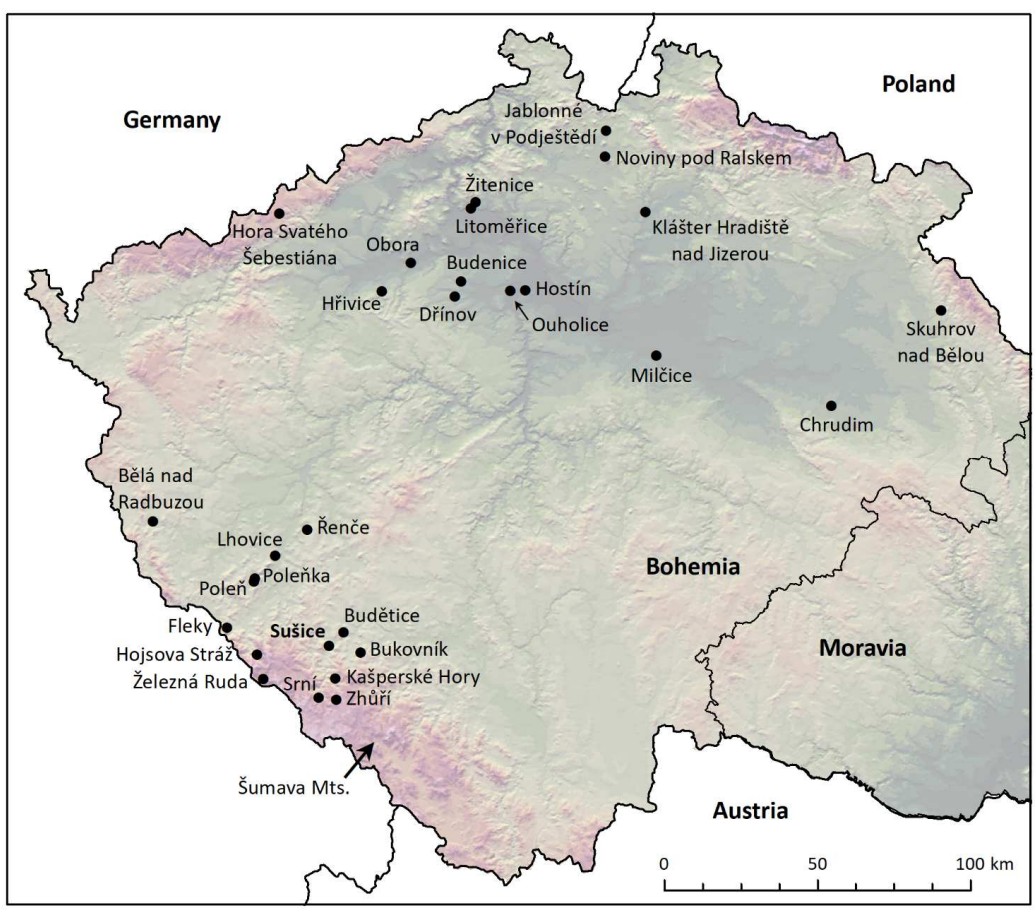

**Figure A1. Location of places over the Czech Republic cited in Sect. 4.4.1.**

**Data availability.** The datasets and series used in this article are available on Brázdil, R., Lhoták, J., Chromá, K., & Dobrovolný, P. (2024). Data from public granaries as a source of proxy data on grain harvests and weather extremes (Sušice region, Czech Republic) [Data set]. Zenodo. https://doi.org/10.5281/zenodo.13918307, last access: 24 October 2024, 2024.


**Supplement.** The supplement related to this article is available online at: https//doi.org/xxxxx.





**Author contributions.** RB designed and wrote the paper with contributions from all co-authors. JL collected all public granary data, interpreted them, and contributed to the paper's writing with historical knowledge. KC performed the basic calculations concerning grain harvests and climate series and finalized all figures. DC contributed expertise on the function of granaries and added to the paper's overall conclusions and approach. PD contributed to the methodology and analysis of the relationship between grain harvests and climate variables. HH contributed to methodology and knowledge of public granaries in Northern Europe. All authors have read and commented on the latest version of the paper.

**Competing interests.** The contact author has declared that none of the authors has any competing interests.

**Special issue statement.** This article is not a part of the special issue. It is not associated with a conference.

**Acknowledgements.** Rudolf Brázdil was supported by the Johannes Amos Comenius Programme and the Ministry of Education, Youth and Sports of the Czech Republic for the project "AdAgriF – Advanced methods of greenhouse gases emission reduction and sequestration in agriculture and forest landscape for climate change mitigation" (CZ.02.01.01/00/22_008/0004635). Dominik Collet acknowledges support from the Centre for Advanced Study (CAS) in Oslo, Norway, and the Norwegian Research Council grant no. 315441 on "ClimateCultures". We acknowledge English language corrections by Laughton Chandler (Charleston, SC).

**Financial support.** This research has been supported by the project "AdAgriF – Advanced methods of greenhouse gases emission reduction and sequestration in agriculture and forest landscape for climate change mitigation" (CZ.02.01.01/00/22_008/0004635).

**Archival sources**

AS1: Archiv Akademie věd České republiky, inv. č. 775 (stará sign. VI C 8, č. 10), Meteorologische Beobachtungen vom Jahre 1808 in Schüttenitz nahe bei Leitmeritz gemacht von Fr. Jac. H. Kreybich.

AS2: Archiv Akademie věd České republiky, inv. č. 777 (stará sign. VI C 8, č. 12), Meteorologische Beobachtungen vom Jahre 1810 in Schüttenitz nahe bei Leitmeritz gemacht von Fr. Jac. H. Kreybich.

AS3: Archiv Akademie věd České republiky, inv. č. 778 (stará sign. VI C 8, č. 13), Meteorologische Beobachtungen vom Jahre 1811 in Schüttenitz nahe bei Leitmeritz gemacht von Fr. Jac. H. Kreybich.

AS4: Arciděkanský farní úřad Kašperské Hory, Farní kronika Srní 1836–1952.

AS5: Arciděkanský farní úřad Kašperské Hory, Pamětní kniha farnosti Kašperské Hory 1726–1959.

AS6: Národní archiv, fond Kontribučenské fondy, inv. č. 571, karton č. 1065–1066; inv. č. 875, karton č. 1574–1575 (Nalžovy).



AS7: Národní archiv, fond Kontribučenské fondy, inv. č. 586, karton č. 1090–1091 (Kolinec).

AS8: Národní archiv, fond Kontribučenské fondy, inv. č. 854, karton č. 1534 (Albrechtice u Sušice).

AS9: Národní archiv, fond Kontribučenské fondy, inv. č. 857, karton č. 1540–1541 (Kašperské Hory).

AS10: Národní archiv, fond Kontribučenské fondy, inv. č. 880, karton č. 1578 (Hlavňovice).

AS11: Národní archiv, fond Kontribučenské fondy, inv. č. 884, karton č. 1586–1587 (Hrádek u Sušice).

AS12: Národní archiv, fond Kontribučenské fondy, inv. č. 890, karton č. 1594 (Kněžice).

AS13: Národní archiv, fond Kontribučenské fondy, inv. č. 895, karton č. 1601–1602 (Dlouhá Ves).

AS14: Národní archiv, fond Kontribučenské fondy, inv. č. 898, karton č. 1606 (Lipová Lhota).

AS15: Národní archiv, fond Kontribučenské fondy, inv. č. 902, karton č. 1610 (Mačice).

AS16: Národní archiv, fond Kontribučenské fondy, inv. č. 913, karton č. 1622 (Palvínov).

AS17: Národní archiv, fond Kontribučenské fondy, inv. č. 921, karton č. 1643–1646 (Žichovice).

AS18: Národní archiv, fond Kontribučenské fondy, inv. č. 924, karton č. 1652–1653 (Sušice).

AS19: Národní archiv, fond Kontribučenské fondy, inv. č. 938, karton č. 1674 (Dolejší Krušec).

AS20: Národní archiv, fond Kontribučenské fondy, inv. č. 940, karton č. 1676 (Dolejší Těšov).

AS21: Národní archiv, fond Kontribučenské fondy, inv. č. 947, karton č. 1687–1688 (Velhartice).

AS22: Národní archiv, fond Kontribučenské fondy, inv. č. 955, karton č. 1703 (Žikov).

AS23: Státní okresní archiv Chrudim, fond Sbírka rukopisů, inv. č. 6: Jan Chlebeček – paměti (před 1849).

AS24: Státní okresní archiv Česká Lípa, fond Sbírka rukopisů, sign. 13/3: Witterungs-Geschichte. Auszug aus den Titl: Lesenswürdige Sammlungen der hinterlegten Jahrgängen. Von Anton Lehmann Lehrer in Neuland. Abgeschrieben durch

Joseph Meißner.

AS25: Státní okresní archiv Domažlice, fond Farní úřad Bělá nad Radbuzou, inv. č. 1, sign. K 1, Pamětní kniha 1786–1938.

AS26: Státní okresní archiv Klatovy, fond Archiv města Poleň (nezpracovaný fond), Kronika obce 1835–1848.

AS27: Státní okresní archiv Klatovy, fond Archiv obce Lhovice, inv. č. 6, sign. K 6, Pamětní kniha 1836–1849.

AS28: Státní okresní archiv Klatovy, fond Archiv obce Poleňka (nezpracovaný fond), Kronika obce 1834–1925.

AS29: Státní okresní archiv Klatovy, fond Farní škola Bukovník, inv. č. 25, sign. K 25, Pamětní kniha 1815–1869.

AS30: Státní okresní archiv Klatovy, fond Farní úřad Budětice (nezpracovaný fond), Pamětní kniha 1649–1981.

AS31: Státní okresní archiv Klatovy, fond Farní úřad Hojsova Stráž (nezpracovaný fond), Pamětní kniha 1820–1946.

AS32: Státní okresní archiv Klatovy, fond Farní úřad Zhůří (nezpracovaný fond), Pamětní kniha 1763–1946.

AS33: Státní okresní archiv Klatovy, fond Farní úřad Železná Ruda (nezpracovaný fond), Pamětní kniha 1787–1933.

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
