# Peer review of "Public granaries as a source of proxy data on grain harvests and weather extremes"

_Climate of the Past, 2024_

## Author Comment (AC1)

This is a well-organized and thorough case study, which examines how granary records may reveal the impacts of climate variability on harvests. To my knowledge, it is the first study of its kind, and the methods developed here could be useful in other regional contexts as well. RESPONSE: We would like to thank the anonymous referee #1 for evaluating of our paper and raising several critical comments, which we are trying to answer below.

Before publication, I would like the authors to address the following:

First, since past studies have usually examined the impacts of climate variability on harvests by looking at grain *prices*, could the authors specifically address what the study of granaries contributes to our understanding of climate/weather and harvests that the study of prices does not? For example, does this study indicate more vulnerability or less vulnerability of farmers' harvests and livelihoods to climate variability than the grain price data indicate? Does it indicate the same or different patterns and trends in climate/weather impacts on harvests? RESPONSE: As follows from many studies (e.g., Camenisch, 2015; Esper et al., 2017; Ljungqvist et al., 2022, 2023; Brázdil et al., 2024), climatic variability can explain only a small portion of grain price variability. This is related to the fact, that besides natural conditions grain prices are influenced also by various societal and socio-economic factors like, for example, wars, administrative decrees, corn reserves, anticipated grain yields, the movement of grain into and out of the country, frequency of grain markets, speculation, etc. (Petráň, 1977). In contrast, data of public granaries indicate direct responses of farmers (serfs) to grain harvest of the preceding or current year. As we state, they show „more vulnerability of farmers' harvests and livelihoods to weather patterns than the grain price data indicate". It means, the granary data did not show the same "patterns and trends in climate/weather impacts on harvests" as price studies do, and they represent a new type of proxies utilisable for the study of poor and abundant grain harvests and their relationships to weather/climatic factors.

Second, the study should discuss (at least briefly) how granaries might have interacted with grain markets or with farmers' behavior. Currently, the study seems to assume that farmers used the granaries only as intended—that is, they borrowed when times were bad and paid back when times were good. But might farmers have used the granaries in other ways? Did they come to rely to some extent on the granaries and take greater risks? Were the grains that were grown from seed corn taken from the granaries used primarily for consumption, or were they sold to buy other food, pay rent, etc.? Could farmers have tried to borrow seed to expand production in expectation of higher prices? Might farmers have willingly maintained a debt to the granary in order to plant or sell more grain and thereby improve their financial condition during average or good years? If there is no evidence that farmers did any of these things, then it would help to add a couple of sentences explaining this. RESPONSE: Accepted, the following text was added as the third paragraph to Section 5.1 as follows:
"The speculative sale of granary grain on the free market was excluded legislatively. According to section 14 of the patent issued in 1788 (Kalousek, 1910), only surplus granary grain beyond annual sowing reserve from stored grain could be sold. This was possible only with the agreement of the state administration (district offices) and the obtained money was saved in dedicated contribution (tax) cash desks or used for granary building. Later it was recommended to sell long stored grain and from the corresponding income to buy new grain with aim to ensure the grain quality. Granary records never report other reasons that led serfs

to borrow grain (only the need for food or seed corn was ever mentioned). Due to this, the associated costs and the fact that granaries never achieved the needed annual sowing reserve, it is unlikely that serfs might have also borrowed grain as a form of speculation instead of coping with actual dearth. In addition, it appears that the local granary accountants had very good knowledge about real situation of serfs."

Third, the visualization of the results (i.e., presentation of the various indices) could be improved for greater clarity and utility. The way results are presented in tables 2 and 3 makes it very difficult to identify the evolution of good and bad years, to identify periods of frequent or consecutive harvest failures, and to compare the performance of different grains to one another. It is also hard to judge the reliability of good/bad harvest determinations based on data from only 1-2 domains. (E.g., how should I compare the third worst year based on only one domains' data with the fourth worst year based on seven domains' data?) Therefore, the ranked lists of the worst and best harvest years for different grains were not especially helpful. RESPONSE: Accepted. Following the reviewer comment and contribute to better visualisation of final results, we prepared a new figure complemented by corresponding text as the first paragraph in Sect. 4.4.2 as follows:
"Table 4 lists the bad and good harvest years separately for rye, barley, and oats in the Sušice region, derived from the calculated weighted grain indices in Tables 2 and 3. The number of bad harvest years was higher than good harvest years for rye (15 vs. 13) and oats (11 vs. 7), while for barley, both datasets were represented by 12 years. Fig. 8 shows some alternation of bad and good harvest years during the study period, lasting no longer than three consecutive years (1806–1808 and 1817–1819 for good harvests and 1825–1827 for the bad harvest). All three considered cereals experienced concurrently good harvests in four years (1808, 1823, 1830, 1838) and bad harvests in seven years (1811, 1812, 1815, 1820, 1821, 1827, 1842). Two cereals had good or bad harvests in four years each, but in 1826 a good harvest of oats and a bad harvest of ray occurred, while in 1847 the situation was reversed. As for a single cereal, good harvests were detected in 10 years compared to seven years with bad harvests."

[Figure]

Figure 8. Fluctuations in good and bad harvest years for rye, barley and oats in the Sušice region during the 1789–1849 CE period.

Concerning of smaller numbers of domains with data available, we respond in lines 249-251 as follows: "Years identified from a higher number of domains indicate greater regional representativeness of poor harvests compared to those derived from a smaller number of domains, which may reflect more localized prominence." We cannot fully agree with statement that "Therefore, the ranked lists of the worst and best harvest years for different grains were not especially helpful." We did not compare the worst and best harvest of the

given year among three cereals used, because (as we show in Fig. 8) their meteorological reasons may differ significantly (see Sect. 4.4.2).

It might have been helpful to see a single timeline showing all index values for all grains each year (or at least whether each index was >1, 0.5–1, -0.5–1, <-1). If such a figure would be too messy, then the authors should determine some other appropriate way to visualize the data so that the evolution of good and bad years, the periods of frequent or consecutive harvest failures, and the comparative performance of different grains is easier to see.
RESPONSE: Accepted. Values of all indices are presented in Figs. 6 and 7, i.e. it seems not to be reasonable to prepare another such figure. To include all index values in a figure would not provide a better visualization. For this reason, we decided to prepare new Figure 8 (see preceding point), which presents in easy form "the evolution of good and bad years, the periods of frequent or consecutive harvest failures, and the comparative performance of different grains."

I would have also liked to see some brief discussion of the correlation among harvests for different grains in the same years, and whether there were lags or autocorrelation in the indices (possibly indicating persistent agricultural problems or economic hardships following especially bad harvest years).
RESPONSE: Correlation analysis "among harvests for different grains in the same years" is discussed in Section 5.2 (see lines 499-508) and summarized in Table 5. Concerning "lags or autocorrelation in the indices", we provided an additional autocorrelation analysis revealing that there is no significant ACF(1) in used indices of bad and good harvests for all three cereals. It indicates no occurrence of persistent periods of bad or good harvest.

Fourth, when discussing the relationship between climate/weather and harvests, the paper considers only those years which had a good or bad harvest and then examines the range of weather conditions for those years. This is useful, but this approach can only investigate one kind of causal relationship: how *necessary* were certain climate/weather conditions for a good or bad harvest [i.e., the probability of good or bad weather given a good or bad harvest, or p(weather|harvest)]. Sometimes, we might want to know how *sufficient* certain climate/weather conditions were for a good or bad harvest instead [i.e., the probability of a good or bad harvest given some good or bad weather, or p (harvest|weather)]. In fact, for some kinds of studies, the question of causal sufficiency will be more important than causal necessity. For example, it would be interesting to know whether or not very cold wet summers consistently brought bad harvests (a question of causal sufficiency), even if most harvests failures might have happened for other reasons (a question of causal necessity). To address causal sufficiency, we would have to start by examining the years with good or bad climate/weather conditions and then see how often they brought good or bad harvests, which is the reverse of the current discussion in section 4.4.2. This should not require a lengthy analysis, but only a test of the most likely patterns, or even some simple counting: e.g., "of *x* summers with below average temperature and above average precipitation, *y* were followed by a poor harvest, as indicated by *z* indices."
RESPONSE: We appreciate these valuable comments and constructive ideas and agree that this would be interesting. However, we have had to take into account that granary and climatic data are biased by many uncertainties, described in detail in Sect. 5.1. As we state in the first sentences: "Various types of uncertainties are present in the granary data due to the multiple and sometimes competing purposes and interests associated with them, as well as their connection to a dynamic agricultural ecology. These factors contribute to some data gaps during the period analyzed." With respect to missing data (see Sect. 4.2 and Fig. 5), our

selection of extreme years in Tables 2 and 3 need not to detect all years of worst and best grain harvests that occurred during the entire analysed 1789–1849 CE period. With respect to these facts and looking on uncertainties in temperature, precipitation and scPDSI reconstructions (see lines 481-485 in Sect. 5.1), addressing "causal sufficiency" would be rather problematic. Moreover, composite analysis in Fig. 8 (newly Fig. 9) shows relatively broad ranges in which temperature, precipitation and scPDSI patterns may change in selected years of bad and good grain harvests. As they are influenced by local natural and weather conditions, they are not well expressed in related reconstructions for the entire Czech Republic.

---

## Author Comment (AC2)

**General comments**

Thank you for the opportunity to review this manuscript. To my knowledge, this is a first in a European (if not global) context and introduces an important new proxy for the documentary analysis of historical grain storage and hence climate variability.

The paper is beautifully written, clearly structured, contextually rich and well supported through references to previous studies. The historical context for the data is recorded clearly and succinctly, and at a level suitable for an international audience. Other than the specific point below, the methodology is clear and straightforward. The results are well presented and explained, and the statistical analysis clear and appropriate.

I really value the transparency over data availability from granaries shown in Figure 5, the corroboration of good and bad harvests from more general documentary evidence in section 4.4.1, and the comparison with wider regional studies. The discussion of the links between grain harvests and climate is strong, reflecting the multiple possible causes of a good vs. bad year. The reflections on the challenges of using grain harvest data are discussed honestly and openly. In short, aside from one relatively minor comment, I have no hesitation in recommending the manuscript for publication.

RESPONSE: We would like to thank the anonymous referee #2 for evaluation of our paper and raising several critical comments, which we are trying to answer below.

**Specific comments**

Lines 168-173. I'm interested to hear more about how well this methodology deals with multi-year (as opposed to single-year) poor harvests. Did multi-year runs of crop failure occur during the study period and, if so, what happened as a result in granaries? I notice at least one instance of this mentioned in the results. There is also mention in Lines 444-446 of granaries being empty – an obvious limitation of the proxy – due to factors in addition to harvest levels.

RESPONSE: Multi-year poor harvests – as follows from Table 4, lasting no more than two years for a given cereal – were reflected in the second year of a bad harvest by minimal quantity of returned grain and, on the contrary, by a repeated borrowing of further grain. On the other hand, single harvest failure did not exclude the restricted return of grain, but there was no reason to borrow grain again in the subsequent year. In other words, consecutive bad harvests were primarily characterized by repeat borrowing of grain.

Line 525. Anthropogenic would be better than man-made.

RESPONSE: Accepted and corrected.

---

## Author Comment (AC3)

**General Comments**

This original study is very interesting because it shows how grain records (rye, barley, and oats in this case) from public granaries may be used as proxies for evaluating grain harvests in relation to weather and climatic patterns. It is a (first) and promising attempt using granary data for historical-climatological research.

This paper is based on a set of unpublished historical granary data coupled with long-term series reconstructed for three basic climatic variables already available (temperature and precipitation series + scPDSI) and well-studied for the Czech Lands territory. In addition, the few existing biases for their use are also explained (ligne 481 to 487).

The choice of data processing (selection of four annual grain variables) statistical analyses (Spearman's rank correlation coefficient) is wise, easy to understand and clearly describe. There's no excess statistical analysis, which is a very good thing.

The use of four different types of weighted grain indices is a good way of getting around the problem of missing data. This methodology could be used as an example when analyzing similar data available throughout Europe, and for other crop types.

In the section "4.4.1 Extreme harvest years and documentary data", for the early 19[th] century, documentary weather data for the selected years of poor grain harvests & of good grain harvests are very impressive and precise! The role of climatic factors in a good or bad harvest is thus easy to identify.

RESPONSE: We would like to thank the anonymous referee #3 for evaluation of our paper and raising several critical comments, which we are trying to answer below.

Grain data from public granaries therefore appear interesting to identifies bad or good grain harvests in relation to weather and climate patterns, but could they also be used to identify other specific environmental factors such as pest insect attacks (e.g. locusts or beetles) in crops? Of course, these phenomena are often influenced by climatic conditions but this aspect is not mentioned in the article.

RESPONSE: Accepted, the following text was complemented beyond the slightly changed sentence on line 479 in Section 5.1 as follows:

"… or reductions in stored grain due to official misappropriation or pests (e.g., insects, mice). Year after year, 1% of stored grain in each granary was subtracted due to pests and storage manipulation. In case of a greater damage to grain, the state administration (district office) assigned purchases of grain at the most favourable market price and from such money to procure fresh grain immediately. Whenever such purchases were recorded in granary records, they were differentiated expressly from borrowed grain due to weather. Substantial loss due to pests and insect attacks was reported in 1796 at the Albrechtice domain ("*A worm spoiled part of stored grain.*" – AS8) or in 1837 at the Kašperské Hory domain ("*473 měřice*, [i.e. 29,085 L] *of rye from the granary was distributed interest-free among serfs due to a worm danger.*" – AS9)."

As for locusts, the last intense locust outbreak in the Czech Lands occurred in 1748-1749, i.e. outside of the period analysed in our paper – see Brázdil, R., Řezníčková, L., Valášek, H., Kiss, A., Kotyza, O. (2014): Past locust outbreaks in the Czech Lands: do they indicate particular climatic patterns? Theoretical and Applied Climatology, 116, 343–357, doi: 10.1007/s00704-013-0950-9, for more details.

Ligne 540 to ligne 552 : This paragraph about other important non-climatic factors for years of bad and good harvests, especially conflicts and wars at that time, is particularly welcome, as it avoids the (potential) criticism of an overly deterministic vision.
RESPONSE: Thank you.

The change in crop type (here the increasing importance of potato growing) as a factor influencing cereal production should be further explored, in connection with the evolution of cultivation methods in the early 19th century. The Industrial Revolution in the 19th century brought technical and technological advances which had an impact on the development of arable farming. Scientific advances, such as mechanisation and artificial fertilizer improved yields.
RESPONSE: Thank you for your constructive comment with respect to potential future research.

Ligne 555 : "*Specifically, it identifies bad or good grain harvests in relation to weather and climate patterns, situating them within the broader context of the Czech Lands in the late 18th and the first half of the 19th century*", it's ok ! But, as the dataset in the article concerns only the Czech Republic, the data is not combined comparatively across wider areas, I'd suggest changing the title slightly from "for historical climatology" to "for historical climatology in Czech Republic".
RESPONSE: We would like to preserve the original title of the paper. The Czech Republic is used here as a case study to demonstrate that this type of data and the proposed methodology of their analysis can be used for historical climatological research in general, irrespective of the country in question – please see our expressions on lines 557-566.

Maps, charts and graphs are clear, well presented and easy to interpret. The English is very good, as is the style.
RESPONSE: Thank you.

The article is perfectly suited for Climate of the Past and deserves to be published with just a few minor revisions.
RESPONSE: Thank you.

**Specific comments (about the references used):**

Various works are cited for different European and Asian countries (China), but for France, only the work of Kaplan in 1977 is cited.
The question of grain harvests and grain management has, however, been discussed at length from various angles in the masterpiece of the French historian Jean Meuvret, published in 1977, "*Le problème des subsistances à l'époque de Louis XIV*" I. La production des céréales dans la France du XVIIe et XVIIIe siècle. & II. Le commerce des grains et la conjoncture (J. Meuvret, Mouton & Cie and École des Hautes Études en Sciences Sociales, Paris, 6 vol.). These works cover much of the 17th and early 18th centuries, and regularly refer to granaries, so they could have been cited. However, the authors may not be familiar with these works, which are unfortunately only available in French, not widely distributed and not easily accessible.
RESPONSE: Thank you for reminding us of these important works. We now cite the papers in the text and references as:

Meuvret, J.: Le problème des subsistances à l'époque de Louis XIV. Tome I: La production des céréales dans la France du XVIIe et XVIIIe siècle. Paris-La Haye et École des Hautes Études en Sciences Sociales, Paris, ISBN 978-2713200342, 224 and 224 pp., 1977.
Meuvret, J.: Le problème des subsistances à l'époque de Louis XIV. Tome II: La production des céréales et la société rurale. École des Hautes Études en Sciences Sociales, Paris, ISBN 978-3110985658, 286 and 275 pp., 1987.
Meuvret, J.: Le problème des subsistances à l'époque de Louis XIV. Tome III: Le commerce des grains et la conjoncture. Éditions de École des Hautes Études en Sciences Sociales, Paris, ISBN 978-2713208867, 191 and 162 pp., 1988.

There's also Abbott P. Usher's book : "The history of the grain trade in France 1400-1710, Cambridge Harvard University Press, 1913" (book in open access), which refers to the granaries, but this French-centric study is now a little outdated, and we prefer to use J. Meuvret.
RESPONSE: We follow your recommendation and we are now citing only Meuvret (1977, 1987, 1988).

**Technical corrections about the bibliography:**

For France, the work of "Gast, M. and Sigaut, F. 1979" appears in the references but not in the text. A correction is therefore necessary.
RESPONSE: Accepted and complemented on line 56.